

# LOCATE v1.0: Numerical Modelling of Floating Marine Debris Dispersion in Coastal Regions Using Nested Hydrodynamic Grids and Parcels v2.4.2

Ivan Hernandez[1], Leidy M. Castro-Rosero[1,3], Manuel Espino[1], and Jose M. Alsina Torrent[1, 2]

[1]Departament d'Enginyeria Civil i Ambiental (DECA), Laboratori d'Enginyeria Marítima (LIM), Universitat Politècnica de Catalunya - BarcelonaTech (UPC), C. Jordi Girona, 1-3, Barcelona, 08034, Catalunya, Spain
[2]Departament d'Enginyeria Gràfica i de Disseny, Universitat Politècnica de Catalunya - BarcelonaTech (UPC), Avinguda Diagonal 647, Barcelona, 08034, Catalunya, Spain
[3]Facultat de Ciències de la Terra, Universitat de Barcelona, Avinguda C. de Martí i Franquès, s/n, Barcelona, 08028, Catalunya, Spain

**Correspondence:** Ivan Hernandez (ivan.hernandez1@upc.edu)

**Abstract.** The transport mechanisms of floating marine debris in coastal zones remain poorly understood, primarily due to the complex geometry and influence of coastal processes which pose difficulties in incorporating them into Lagrangian numerical models at coastal scales. The numerical model presented in this study addresses these challenges using nested grids of varying hydrodynamic resolutions (2.5 km, 350 m, and 70 m). The model couples Eulerian hydrodynamic data with a Lagrangian

particle solver to accurately simulate the motion of plastic particles in coastal regions. A Lagrangian validation of the model was conducted using drifter data to assess the model's skill and confidence in simulation predictions. The results demonstrated high skill and stability in 72-hour forecast horizons and high skill score values close to the shoreline. Additionally, a beaching sensitivity analysis was performed to determine particle beaching parameterisations suitable for coastal zones. This analysis demonstrated that the real-time distance of particles to the shore during simulations was the most accurate method for detecting

the land-water boundary at coastal scales, accurately representing residence times and spatiotemporal beaching patterns. Simulations were conducted using nested grids and a single coarse grid, both using the same plastic input from river outflows in a pilot test at the Barcelona coastline with a complex morphology due to coastal structures, two major river mouths, and a large harbour. The simulations revealed marked differences in beaching amounts and residence times. Both simulations exhibited high levels of beaching, with the nested grid simulation registering 91.5% beaching and the low-resolution grid simulation

showing 95.8% beaching, surpassing other studies conducted at larger scales. Beaching amounts exhibited high levels of variability within demarcated areas between simulations, with the highest flux observed near the Llobregat River mouth release point. The inner port area showed that particles have 18 times longer residence times when using high-resolution data, demonstrating the model's ability to solve complex coastal geometrical structures. By using nested grids, the model can resolve coastal processes with high-resolution hydrodynamic data and accurately simulate potential accumulation zones and litter hot-spots at

coastal scales.





# 1 Introduction

## 1.1 Marine debris in coastal regions

Coastal regions are highly susceptible to the impacts of the presence of marine debris (also widely referred to as marine litter), affecting ecological resources, social activities and economic assets. (Browne et al., 2015). River discharges are widely acknowledged as being an important vector for the transport of debris from land-based sources to coastal areas (Galgani et al., 2015; Lebreton et al., 2017; Rech et al., 2014). Near the shoreline, coastal transport processes are crucial in determining marine debris' residence times and accumulation zones. In coastal zones, debris can accumulate, return to the emerging beach, sink to the seafloor, or migrate to the open sea where it can converge in oceanic accumulation regions such as subtropical gyres as predicted by Ekman dynamics (Canals et al., 2021; Eriksen et al., 2014; Ioakeimidis et al., 2019; Law et al., 2010; Lebreton et al., 2012, 2018, 2019; Liubartseva et al., 2018; Maximenko et al., 2012; Ryan, 2015; van Sebille et al., 2015; Woods et al., 2021). In contrast with findings in oceanic waters, numerical modelling studies show that debris accumulation in open waters of the Mediterranean Sea is unlikely, with 49%-63% accumulating in coastal regions and the remainder sinking to the seafloor (Kaandorp et al., 2020; Liubartseva et al., 2018). Marine debris can undergo repeated episodes of stranding and release from the shoreline, as well as deposition and resurfacing on the coastal seabed, with a fraction being exported to offshore waters (Hinata et al., 2017; Lebreton et al., 2019; Liubartseva et al., 2018). Smaller fragments are more frequently observed in open water accumulation zones, whereas larger items are more prevalent in coastal regions (Morales-Caselles et al., 2021). While mesoscale circulation transport mechanisms in the open ocean are relatively well understood, our knowledge of the motion of plastic particles in coastal regions, beaching (particle interaction with the beach morphology returning to the shoreline), accumulation or exchange with the open sea is more diffuse (van Sebille et al., 2020).

Coastal hydrodynamic processes occur within a narrow region between 50 m to 600 m from the shoreline and can be very energetic depending on wave energy and local bathymetry. Coastal currents are driven by various forces, including tides, density gradients, storm surges during severe weather events, wind-driven circulation and processes derived from wave action such as rip currents, return bed flow (undertow) and longshore drift. The nonlinearity of ocean waves induces the transport of marine debris in the direction of wave propagation due to Stokes drift, which can be described as the difference between the average Lagrangian and average Eulerian velocities at mean depth (Röhrs et al., 2012; Stokes, 2009). The magnitude of the effect of Stokes drift on a particle is dependent on the buoyancy ratio between the particle and seawater (Alsina et al., 2020; Chassignet et al., 2021; Chubarenko et al., 2016). The role of Stokes drift, however, is dampened with depth, with implications for particles that sit lower in the water column. Stokes drift is widely acknowledged as being one of the key components for numerical simulations of floating marine debris drift.

## 1.2 Beaching of marine debris

The movement of marine debris on the shore is predominantly influenced by wave energy and direction, with a tendency for alongshore transport concentrating in convergence zones before being backwashed offshore by nearshore currents (Kataoka and Hinata, 2015). The residence time of debris at sea is dependent on the buoyancy ratio and the upward terminal velocity of





the particle (Hinata et al., 2017; Isobe et al., 2014; Yoon et al., 2010). Larger debris items possess a higher upward terminal
velocity determined by size and specific gravity. These tend to drift on the uppermost layers and not sink from entrapment
by eddies under breaking waves, thus being pushed further onto the upper backshore by swash waves which have a positive
effect on beach residence times before being moved again (Hinata et al., 2017). Smaller debris items move upward due to
buoyancy and intense turbulent mixing but are subject to greater friction by seawater (Hinata et al., 2017; Isobe et al., 2014).
Longer residence times on beaches allow for increased fragmentation, yielding microparticles that are subsequently washed
back onshore by wind and wave action since degradation by ultraviolet radiation is severely hindered when floating on seawater
(Andrady, 2011; Isobe et al., 2014). A natural sorting largely responsible for the removal of debris from the upper ocean surface
layer has been hypothesised in coastal environments with debris potentially exposed to repeated cycles of beaching, settling and
resuspension (Koelmans et al., 2017; Lebreton et al., 2019; Morales-Caselles et al., 2021). The influence of coastal processes
on the beaching of virtual particles in simulations is still mostly unexplored (Hinata et al., 2017).

## 1.3 Modelling transport of marine debris in coastal waters

The majority of studies that use Lagrangian models to track the dispersion of plastic particles do not resolve at coastal scales
or do not use hydrodynamic inputs that consider the complexities of coastal transport processes (van Sebille et al., 2020). Due
to the longevity and relatively high buoyancy of plastic particles, the residence times in coastal environments can be large,
potentially travelling great distances (Maximenko et al., 2012). Consequently, coastal numerical approaches that focus on
small scales with a high spatial discretisation but a small spatial domain will experience plastic particles moving outside of the
domain boundaries relatively fast, especially in energetic conditions. These challenges are also related to the modelling of the
relevant coastal processes from a hydrodynamic perspective (Critchell and Lambrechts, 2016). The Lagrangian connectivity of
nearshore flows strongly depends on the horizontal resolution of the underlying Eulerian hydrodynamic data (Dauhajre et al.,
2019).

Accurately simulating marine debris dispersion in coastal regions is challenging due to difficulties in simulating coastal
processes, such as wave breaking, wave-induced currents and coastal currents. Additionally, small computational meshes are
required for coastal processes at smaller and varying spatial scales than for oceanic processes. Small computational grid sizes
are also needed to characterise the influence of the complex shoreline configurations such as dykes, beaches and harbours in the
motion of virtual particles, as well as in the hydrodynamic-topography interaction. Coastal numerical simulations require a high
spatial coverage to simulate particle exchange between regions and a high spatial resolution close to coastlines to solve coastal
processes. Conducting simulations at a high resolution over large geographical areas is technically difficult as it increases the
computational costs exponentially due to the increase in the total number of computational points (nodes) in the domain with
a reduced grid size. Nested grid domains can cover relatively large areas around the coastline with lower resolutions yet have
smaller grid subsections with higher resolutions on coastal regions of interest where the coastal processes and the topography
demand it. This approach can overcome the spatial limitations of higher resolutions required to simulate coastal processes
while allowing the movement of virtual particles across different mesh domains.



## 1.4 Objectives

The goal of the present work is to simulate particles taking into account coastal processes using nested hydrodynamic grids at varying resolutions. A numerical model was developed to consider coastal processes at a spatial resolution according to coastal process variability. The numerical model Prediction of pLastic hOtspots in Coastal regions using sATellite derived plastic detection, cleaning data and numErical simulations in a coupled system (referred to as LOCATE hereon) is made of two sub-modules:

a. Hydrodynamic module: computes the waves and currents that transport the marine debris using different meshes with varying grid resolutions depending on the domain and applicability.

b. Dispersion module: computes the motion of virtual debris particles within the different computation domains using nested hydrodynamic information at various resolutions, applying the open-source Lagrangian particle solver Parcels, Probably a Really Efficient Lagrangian Simulator, for the simulation of particles. (Delandmeter and Van Sebille, 2019).

For the present work, the Barcelona coastline was chosen to conduct the simulations for the development of the model. The Barcelona metropolitan area is densely populated and the coastline has been identified in several studies as an accumulation hotspot for floating marine debris with a high flux of debris from the coastal regions to open waters. (Liubartseva et al., 2018; Sánchez-Vidal et al., 2021; Zambianchi et al., 2017). Two rivers have estuaries close to Barcelona; the Llobregat River which has a basin area of approximately 5,000 km$^2$, has its estuary adjacent to the south of the Barcelona port, and the Besòs River, with a basin area of 1,000 km$^2$ flows out to sea several kilometres to the north of the city. The availability of high-resolution hydrodynamic data for the region that covered both river mouths was key in its selection for the development of LOCATE, as well as the availability of river outflow data that enabled the simulation of particles based on observational data.

This paper is structured as follows: Firstly, the hydrodynamic data and input data are described, as well as an outline of the Lagrangian particle solver used to conduct simulations. Subsequently, a Lagrangian validation of LOCATE is performed using drifter data in the study domain. The observed and simulated trajectories are compared using the skill score statistical analysis to assess the model's reliability in predicting the dispersion of floating marine debris in coastal regions. Parameterisations for beaching are outlined in a beaching sensitivity test to determine the most suitable particle beaching definition at small scales while considering coastal processes. Additionally, a comparison is made between using nested grids and a single low-resolution hydrodynamic grid for a marine debris simulation. Lastly, a comprehensive discussion is provided describing the advantages and limitations of different parameters in the beaching module, as well as the results from the use of nested grids with future areas of development.





## 2 Method

### 2.1 Installation and configuration of LOCATE

The LOCATE model is built upon the coupling of Eulerian hydrodynamic information and the Lagrangian simulation of marine debris particles using the hydrodynamic information. Parcels is a Lagrangian particle solver that allows user customisation of the different Python tools available therein to produce simulations of virtual plastic particle movement in space and time (Van Sebille et al., 2023). To use LOCATE, the creation of a new Python environment is recommended, within which the necessary Python libraries can be installed using the `requirements.txt` script, for which full instructions are provided in the LOCATE repository (Hernandez et al., 2023). Parcels and its dependencies can then be installed into the newly created LOCATE environment using the appropriate `*.yml` file for the user's operating system. A simulation configuration file is found inside the `config` folder. General variables are set in this file, such as a simulation identifier, study domain coordinates, whether a simulation writes new data or uses existing data, and various forms of plotting outputs, such as particle trajectory paths, concentration maps, or animations.

### 2.2 Eulerian hydrodynamic information

The premise behind LOCATE is the use of nested hydrodynamic grids of varying resolutions for coastal-scale marine debris simulations, obtained from a circulation model coupled with data from a wave propagation model. These data can be selectively downloaded as required, with the download period and the directories from which the simulation will run specified in the configuration file. Downloading the necessary files can be done using the `Download_data` script.

The circulation numerical model used for the lower resolution regional scale simulations has a horizontal resolution of 1/36°, or ~2.5 km and is based on the Nucleus for European Modelling of the Ocean (NEMO) numerical model for the Copernicus Marine Environment Monitoring System (CMEMS) that became fully operational in May 2015 (CMEMS, 2023; Gurvan et al., 2017). The Iberian Biscay Irish (IBI-CMEMS) system encompasses the Atlantic and Mediterranean regions of the Iberian peninsula and is forced with 3-hourly atmospheric fields provided by the European Centre of Medium Weather Forecast (ECMWF) (Sotillo et al., 2015). Hourly pre-computed simulations were downloaded from the CMEMS website, and the operational Iberia Biscay Irish – Monitoring Forecasting Centre (IBI MFC) analysis and forecasting system of CMEMS was used that provides daily model estimates and 5-day forecasts of various physical parameters (Sotillo et al., 2015, 2020). The IBI products on CMEMS have been extensively validated and can be used with confidence to characterise circulation and regional and oceanic scales, although limitations have been observed at smaller coastal scales (Sotillo et al., 2015, 2020). To bridge the gap between larger regional services and end users that require high-resolution data for smaller scales such as harbour environments, information from CMEMS is downscaled by downstreaming services to adequately represent air-sea interactions involving atmospheric-wave-ocean coupling (García-León et al., 2022). User credentials are required to download data from CMEMS which can be included as environment variables (`CMEMS_USER` and `CMEMS_PASSWD`) or be placed in a `.env` file in the root LOCATE folder.





The *Sistema de Apoyo Meteorológico y Oceanográfico a las Autoridades portuarias* - System of Meteorological and Oceanographic Support for Port Authorities (SAMOA) provides fully customised ocean-meteorological information to a multitude of Spanish port authorities consisting of several modules; from near real-time observational capabilities to local high-resolution

forecast modelling for atmosphere, waves and ocean circulation. The SAMOA system has been available from *Puertos del Estado* - Spanish Port Authority (PdE) since January 2017 and provides an integrated coastal and harbour forecast service of sea-level, circulation, temperature and salinity fields, that in the case of Barcelona includes forcing due to freshwater discharges from rivers using climatological data and a constant salinity of 18 PSU (García-León et al., 2022; Sotillo et al., 2020).

The SAMOA model application incorporates two regular grids; a coastal grid with a spatial resolution of 350 m and a

harbour grid with a spatial resolution of 70 m nested into the coastal grid from a computational cluster property in the PdE website using the Open-source Project for a Network Data Access Protocol (OPeNDAP) (OPeNDAP, 2022). The five-fold nesting ratio between grids has a sufficient definition to reproduce circulation within harbours due to their inner shape while considering larger scale dynamics of the coastal domain (Sotillo et al., 2020). Coastal and harbour grids use the numerical model based on the Regional Ocean Modelling System (ROMS) and data are calculated daily using coastal simulations using

data from metocean operational products nested into the IBI-CMEMS forecast solution using the SAMOA system (Alvarez Fanjul et al., 2018; García-León et al., 2022; ROMS; Shchepetkin and McWilliams, 2005; Sotillo et al., 2015). The nested SAMOA models are driven by sea surface data forced by the AEMET HARMONIE 2.5 km model (wind stress, atmospheric pressure, surface heat and water fluxes) with a 36-hour forecast horizon and a prior land mask applied to the forcing data to avoid land contamination.

The harbour domain grid uses metocean forcing to downscale the coastal domain information to the more detailed resolution required. The nesting strategy was verified for inconsistencies with some continuity found between the IBI-CMEMS data and the SAMOA coastal application boundaries. The overall performance of SAMOA has also been successfully validated and was shown to capture major synoptic and mesoscale features mainly inherited from the IBI-CMEMS solution, together with specific local features (Sotillo et al., 2020). Further validation of the IBI and SAMOA systems was carried out using data

from the extreme Storm Gloria event in January 2020, with both systems capturing the arrival of the storm with adequate accuracy (Sotillo et al., 2021). There are no user credentials required to download data from the OPeNDAP server, although the downlaod script is configured to download data at coastal and harbour resolutions for the Barcelona area. The hydrodynamic grid characteristics can be found in Table 1.

The wave component was obtained using the open-source Simulating Waves Nearshore (SWAN) model adapted to provide

a wave height forecast downscaled to the Barcelona coastline using the nesting grid scheme (Allard et al., 2002). The SWAN-based system is formulated on the spectral reconstruction technique of sea states to compute random, short-crested wind-generated waves in coastal regions and inland waters. The wave data are also downloaded from the CMEMS server using the download script.



**Table 1.** Basic characteristics of the circulation numerical models used in the nested system (IBI, coastal and harbour domains)

|  | Regional domain model CMEMS-IBI | Coastal domain model | Harbour domain model |
|---|---|---|---|
| Model | NEMO-3.6 | ROMS | ROMS |
| Resolution | 1/36° | 350 m | 70 m |
| Depth levels | 50 (unevenly distributed) | 20 $\sigma$ coordinates levels | 15 $\sigma$ coordinates levels |
| Atmospheric forcing | ECMWF (3h) | AEMET HARMONIE 2.5 km (1h) | AEMET HARMONIE 2.5 km (1h) |
| Open boundary conditions | CMEMS GLOBAL (daily 3-D) | CMEMS-IBI (daily 3-D $T$, $S$ and hourly surface currents, sea level and barotropic contribution) | Coastal domain model |

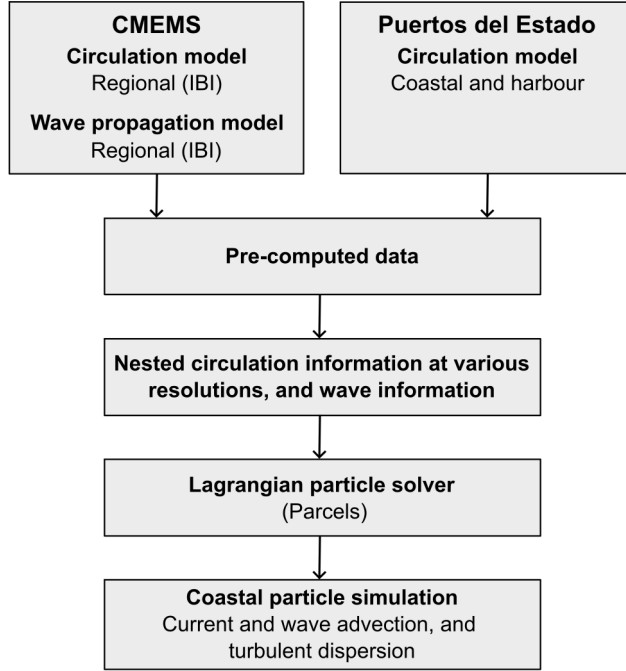

**Figure 1.** Schematic representation of the coupled plastic dispersion model components.



## 2.3 Nested hydrodynamic grids


Circulation data as numerical simulations available from the PdE website were interpolated from a C-grid domain to a regular grid for the coastal and harbour grid domains. This was achieved using the `UPC_resample_datasets` script which carries out an interpolation between the three grids filling out empty values in the grids with higher resolution with the equivalent spatiotemporal data from lower resolution grids in these points. An additional interpolation was performed in the lower resolution

regional grid due to the difference in temporal data since CMEMS data is provided 30 minutes past the hour and OPeNDAP data is provided on the hour, requiring previous and post-simulation temporal data to be downloaded in a process automated in the script.

The $u$ and $v$ components of the hydrodynamic data of each grid were nested using the Parcels `NestedField` object in order of higher to lower spatial resolution, prioritising the order in which data was to be interpolated, as shown in Fig.1,

(Delandmeter and Van Sebille, 2019). This nesting approach allows the highest resolution data to be used in the areas where it is available while utilising lower resolution data elsewhere. The mean module intensity and current velocities in the nested grids (Fig.2a), as well as the coastal and harbour grid boundaries (Fig.2b and Fig.2c respectively), are shown for a specified date. To use nested grids in the simulation, the `nested_domain` variable in the configuration file must be set, otherwise, it defaults to the lowest resolution grid.

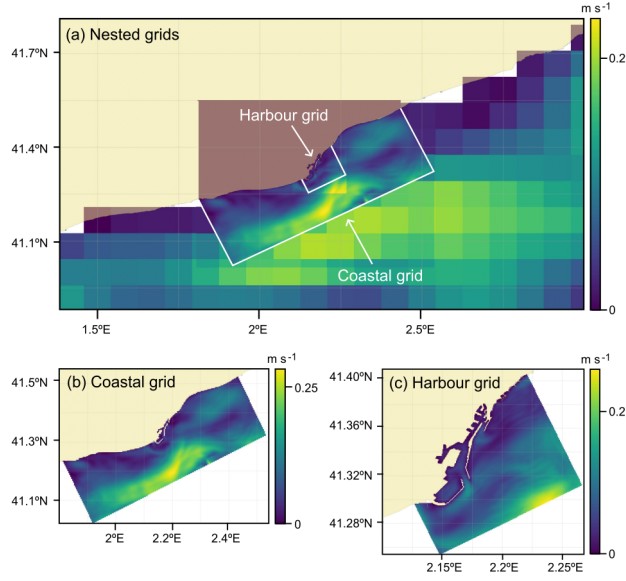

**Figure 2.** Example of hydrodynamic grids displaying module intensity and current velocity (m s$^{-1}$) for 23 March 2017.





## 2.4 Lagrangian particle simulations


The particle dispersion submodel was adapted to work at a coastal scale using the described hydrodynamic information to produce particle trajectory computations of virtual particles. The transport processes included in the motion of the virtual particles are coastal currents, wave data in terms of Stokes drift, turbulent diffusion due to typical stochasticity, and beaching. Particle simulations are conducted using the `UPC_main_simulation` script, with the required functions stored in an objects

file, `UPC_parcels_objects`. If the `nested_domain` variable is set, the `get_nested_fieldset` function in the objects file is assigned as a `fieldset`, a class that holds hydrodynamic data needed to execute particles. Otherwise, the lowest resolution `fieldset`, `get_regional_fieldset` is selected. The setting of the directories for the hydrodynamic data required for each `fieldset` can be found in the configuration file. Wave data is also added into a `fieldset` using a `set_Stokes` function found in the objects file, which reads the $u$ and $v$ Stokes drift components from the CMEMS wave

data.

The particle trajectory simulation starts from an input of particles given at an initial time instant and the position of the particles is then tracked in time moved by the mentioned transport processes. The input of particles can be given by direct information of marine debris particles measured in the domain but it can also be obtained indirectly by inferring information about the amount of debris coming from rivers or wastewater discharge points located within the computational domain (see section 2.8).

These data are included in the simulation file through a `sampler` variable, which in turn is set by a `ParticleXLSSampler` function in the objects file that reads a spreadsheet with river input data. These data assume a continuous hourly or daily release of particles from a predetermined number of points as set in the `particle_frequency` variable in the configuration file. The spreadsheet name(s) and release coordinates are also specified in the configuration file. If the `particle_frequency` variable is set to `None`, the `ParticleXLSCreator` function is used which creates individual, one-time releases for which

spatial coordinates must also be provided in the spreadsheet input data.

The fundamental concept behind Parcels and any Lagrangian analysis is to integrate the advection equation (van Sebille et al., 2018):

$$\frac{d\vec{x}}{dt} = \vec{u}(\vec{x}, t) \tag{1}$$

where $\vec{x}$ is the three-dimensional position of a virtual plastic particle in space, $t$ is the time and $\vec{u}(\vec{x}, t)$ is the three-dimensional

Eulerian flow velocity field. $\vec{u}(\vec{x}, t)$ incorporates the net current velocity and Stokes drift. By integrating both sides of the equation, it can be rewritten as a pathway equation:

$$\vec{x}(t) = \vec{x}(t_0) + \int_{t_0}^{t} \vec{u}(\vec{x}, t) dt \tag{2}$$

which highlights that the location of a plastic particle at time $t$ depends both on the initial three-dimensional position of the particle $\vec{x}(t_0)$ and the velocity field $\vec{u}(\vec{x}, t)$. The pathway equation (2) by itself gives a trajectory of completely passive tracers.

Note that $\vec{u}(\vec{x}, t)$ is by definition a mean current and does not account for particle dispersion induced by diffusivity at smaller temporal scales.





## 2.5 Plastic particle dispersion

In equation (1) the velocity $\vec{u}(\vec{x},t)$ is by definition a mean velocity not incorporating sub-grid scale dispersion processes that are often parameterised as a diffusive process. In Lagrangian particle simulations, diffusive processes can be modelled as a stochastic, random displacement of particle positions as a function of the local eddy diffusivity (van Sebille et al., 2018). Whereas the time evolution of a trajectory advected by the mean current is accurately solved using an Ordinary Differential Equation (ODE), which in the Parcels model is numerically solved using a 4th order Runge-Kutta scheme; the time-evolution of a stochastic process is solved using a Stochastic Differential Equation (SDE). The SDE for a particle trajectory including diffusion is:

$$d\vec{x}(t) = (\vec{u}(\vec{x},t) + K(\vec{x},t))dt + V(\vec{x},t)dW(\vec{x},t)$$
$$\vec{x}(t_0) = \vec{x}_0 \tag{3}$$

where $\vec{x}$ is the particle position vector ($\vec{x}_0$ being the initial position), $\vec{u}$ the velocity vector, $K$ is the diffusivity tensor where $K = \frac{1}{2}VV^T$, and $dW(t)$ is a Wiener increment. The diffusivity tensor is a three-dimensional tensor incorporating the ocean-coastal diffusivity in the three-dimensional space. In Parcels, however, only horizontal diffusivity vectors can be simulated. These can be obtained from experimental data from the numerically simulated Eulerian eddy diffusivity (Bezerra et al., 1997). Three-dimensional eddy diffusivity can be obtained from ROMS simulations as a parameterisation of the sub-grid diffusive processes in accordance with hydrodynamic forcing (wind, density, waves). However, pre-computed data available in the OPeNDAP do not incorporate such information and a constant diffusivity parameter is generally used instead (OPeNDAP, 2022). When the horizontal diffusion coefficient ($K$) is constant and time-space is invariant, equations (3) can be simplified and integrated as:

$$\vec{x}(t) = \vec{x}(t_0) + \int_{t_0}^{t} \vec{u}(\vec{x},t)dt + R\sqrt{\frac{2dtK}{r}} \tag{4}$$

where $R$ is a random process representing subgrid motion with a zero mean and variance $r = \frac{1}{3}$. The hydrodynamic data input assumes current velocity data and Stokes drift data from the wave propagation model. A diffusion parameter was incorporated into the model using Parcels' `fieldset` functionality in the simulation file, by adding into it the `add_constant_field` wrapper function that adds field data that is constant in space. For all simulations described herein, the diffusion parameter value was set to 10 m²s⁻¹, set by the `kh` variable found in the configuration file, as previous tests showed no significant difference with higher values.

## 2.6 Plastic particle behaviours

Parcels allows the creation and use of specific kernels, which are small snippets of code that define particle dynamics and run during the execution of the simulation, depending on the requirements of the simulation (Delandmeter and Van Sebille, 2019). Particle behaviours can be declared as variables within the `PlasticParticle` class in the simulation file, and can





then be associated with a kernel in the `UPC_Parcels_kernels` file, resulting in the output of these data values from the simulation for further analysis. Examples of `PlasticParticle` variables used in kernels are particle residence time (`age`), trajectory length (`distance_trajectory`), real-time distance to the shore (`distance_shore`), and particle beaching (`beached`). The beaching definitions described in section 2.9 used the `beached` variable and different kernels for different

particle dynamics. When particles became beached or were exported from the study domain, they were deleted from the simulation using Parcels' core `DeleteParticle` behaviour.

## 2.7    Lagrangian model validation

A comparison was made between a real trajectory and a simulated trajectory to conduct a Lagrangian validation of the LO-CATE model and evaluate its accuracy in predicting trajectories in coastal regions. Drifter data were used from the Mobile

Autonomous Oceanographic Systems (MAOS) Argo Italy database, available from the National Institute of Oceanography and Applied Geophysics website (OGS, 2023). Drifter data were selected based on not having a drogue to assess only the influence of surface currents as it is assumed more realistic for a floating particle. Drifters that had trajectories that crossed the study domain were selected from the database, with priority given to drifters with data where the high-resolution hydrodynamic grids applied and that were deployed from 2017 for which these data were available. Validation simulations were conducted

for the available drifters. CODE drifter 6592, deployed in February 2022, was chosen as the most suitable because its trajectory crossed the coastal and harbour grids for a period long enough to analyse with a 72-hour horizon. Other available drifters crossed the coastal domain for an insufficient period for such time-horizon analyses and were not selected for the validation results presentation.

     Simulations were conducted from the point where the selected drifter's trajectory transected the coastal grid boundary, at

coordinates 41.18162°N and 2.24084°E. The provided data did not include timestamps, only a deployed and end date, so the drifter trajectory was temporally interpolated to provide hourly data points where 100 particles were released at every step. Particles were released between the period 9 March 2022 18:11:00 to 14 March 2022 18:11:00. The number of particles was determined through a sensitivity analysis of the standard deviation using varying amounts of particles (Castro-Rosero et al., 2023). The same simulations were conducted using nested grids and using the IBI-CMEMS grid only.

The comparison of the simulated versus the observed trajectories is based on the Normalized Cumulative Lagrangian Separation (NCLS) distance Skill Score (SS) tests developed by Liu and Weisberg (2011). NCLS is defined as the cumulative sum of the separation distance between the observed and simulated trajectories (D) weighted by the length of the cumulative observed trajectory length (L):

$$NCLS = \frac{D}{L} = \frac{\sum_{i=1}^{N} d_i}{\sum_{i=1}^{N} lo_i} \qquad (5)$$

where $N$ is the total number of time-steps, $d_i$ is the separation distance between the simulated and observed endpoints at time-step $i$, and $lo_i$ is the length of the observed trajectory. A reduced NCLS value implies an improvement in model performance with a 0 value implying a perfect fit between the simulated and observed values. The SS is calculated from the cumulative $d$ and $lo$ values for all time-steps using a non-dimensional tolerance threshold ($n$) corresponding to the no-skill criterion of the





simulated trajectory. In the same way as the present work, the value of this threshold is generally set to 1 (Liu and Weisberg,
2011; Révelard et al., 2021). The SS is defined as:

$$SS = \begin{cases} 1 - \frac{NCLS}{n}, & (NCLS \leq n) \\ 0, & (NCLS > n) \end{cases} \tag{6}$$

Using the established threshold value, a SS value of 1 implies exact matches between simulated and observed trajectories.
Distances were calculated using the `geopy.distance` Python module using the WGS-84 ellipsoid (Geopy, 2022). As part
of the default particle behaviour in the model, particles that moved out of the study domain boundary (i.e. were exported) or
became beached were detected, recorded, and subsequently deleted from the domain for computational efficiency purposes.
To maintain statistical rigour for the validation analysis, however, particles were not deleted when they became beached or
were exported. Therefore, the same number of particles as were released were used in the validation calculations, with the last
available coordinates registered either on the perimeter of the study domain or on the coastline used to calculate cumulative
distances and the SS for subsequent time-steps.

## 2.8 Plastic particle data inputs and release points

The high amounts of coastal marine debris and high flux rates observed in the study area allow for the testing of beaching
definitions and the applicability of those used in larger-scale studies to smaller coastal settings. Two Lagrangian simulations
were conducted using the same input data; one using nested hydrodynamic grids and the other using a low-resolution IBI-
CMEMS grid to compare particle beaching, residence times, and trajectory distance. Particle release coordinates were selected
based on the most proximal nodes from the coastal grid to each river mouth midpoint since it provided the highest resolution
covering both river mouths. The release point for the Llobregat River (41.294468°N, 2.140995°E) was approximately 180 m
distance from the river mouth midpoint, and that of the Besòs River (41.418909°N, 2.232839°E) was approximately 240 m
from the river mouth midpoint. The selected coordinates for the particle release points are highlighted in bold in Table 2. The
separation distance of both points was within the spatial resolution of the coastal grid.


The coastline in the study domain was divided into 16 different zones as shown in Fig. 3 which also shows the coastal
and harbour grid boundaries. Demarcated zones were based on current official municipal demarcations according to the *Área
Metropolitana de Barcelona* (AMB) or prominent features, such as port areas or beaches.





**Table 2.** Closest nodes from each domain to the river mouth midpoint coordinates showing respective distances. The selected nodes are shown in bold.

| River | Grid | Distance (km) | Latitude (°N) | Longitude (°E) |
|---|---|---|---|---|
| Llobregat | Harbour | 0.03 | 41.294224 | 2.141092 |
| **Llobregat** | **Coastal** | **0.18** | **41.29285** | **2.14149** |
| Llobregat | Regional | 4.82 | 41.333332 | 2.166666 |
| Besòs | Harbour | 1.32 | 41.407047 | 2.232609 |
| **Besòs** | **Coastal** | **0.24** | **41.417671** | **2.235227** |
| Besòs | Regional | 1.46 | 41.416668 | 2.249999 |

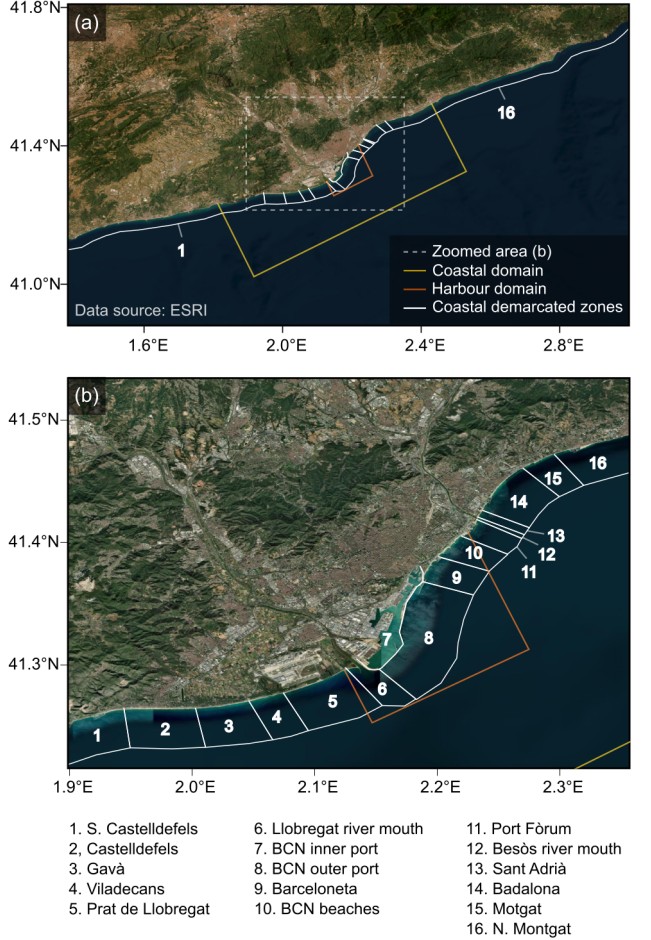

**Figure 3.** Demarcation of zones on the coastline (numbered) within the study domain area (a) and a closeup of the city area (b).



The simulation period was limited to 261 days spanning between 1 February 2017 and 19 October 2017 due to constraints
in the availability of high-resolution hydrodynamic data before February 2017. The input component $\vec{x}(t_0)$ consisted of obser-
vational data of debris outflow from the Llobregat River and Besòs River around the city of Barcelona sourced from Schirinzi
et al. (2020). As the river debris outflow data were irregular in time, a linear interpolation was applied to reflect real-life con-
ditions of continuous debris release throughout the simulation period, as illustrated in Fig.4. A total of 552,480 particles were
simulated throughout the simulation period, with particles being released every hour based on the interpolated daily amounts
from each release point.

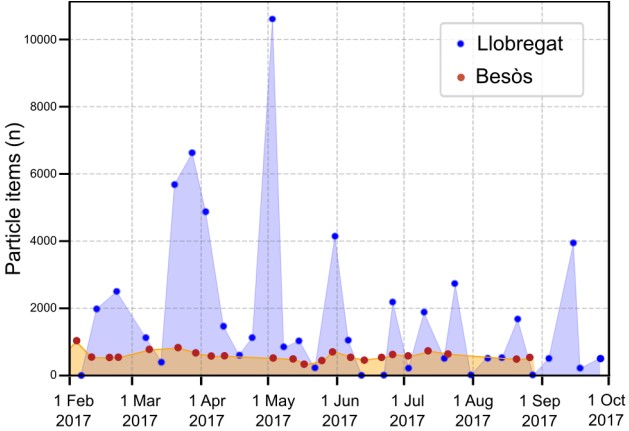

**Figure 4.** Linear interpolation of debris outflow from the Llobregat and Besòs Rivers for the period 1 February to 30 September 2017.
Original data points are represented by dots in the plots.

### 2.9   Particle beaching and beaching sensitivity analysis

A beaching module was included to parameterise particles that crossed the land-water boundary using the variables specified
in section 2.6 and the kernels listed below. A beaching sensitivity analysis was carried out in two parts with three different
scenarios:

a. Using the two release points of river outflow from the Llobregat and Besòs Rivers as per the simulations that compared
   the use of regional and nested grids (Table 2).

   b. Releasing particles homogeneously in the study domain as a control using the coordinates of 132 nodes on the regional
   domain that were on water (Appendix B).

In scenario 1 a particle was considered beached when $u$ and $v$ velocities were effectively stationary ($\leq$1e-16 m$^{-1}$), using
the `beaching_velocity` kernel. In scenario 2 a particle was considered beached when it physically crossed the land-
water boundary determined by a distance-to-shore parameter $\leq$0 m, using the `beaching_distance` kernel. In scenario
3, a time dependency was introduced linked to a distance parameter from the shoreline derived from the daily mean current





velocity of 0.078 m s⁻¹ during the study period, as depicted in Fig. 5. Given the proximity to the shoreline of the particle release points, a time-frame of 6 hours was used resulting in a distance parameter of 1.694 km. The latter scenario used the

`beaching_proximity` kernel.

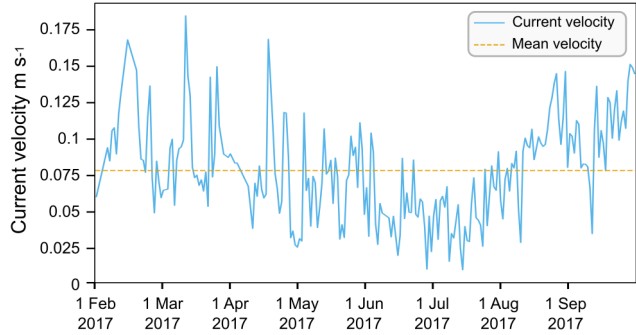

**Figure 5.** Mean daily current velocity of the IBI-CMEMS grid for the period 1 February 2017 to 30 September 2017, with the mean velocity for the period.

Given that kernels in Parcels are limited to basic arithmetic operations and conditions, the Parcel's interpolation capabilities were utilised to calculate the real-time minimum distance between particle and shoreline for scenarios 2 and 3. This was achieved by incorporating the distance-to-shore data into grids with the same structure as the hydrodynamic data. Information on the Catalan coastline was obtained as a linestring in a shapefile format with 259,080 data points, excluding islands or

islets, using UTM-31N with Datum ETRS89 as the cartographic reference system using a scale of 1:50000 (ICGC, 2020). The linestring was edited in the open-source QGIS software to only include data within the boundaries of the study domain. Node coordinates of the hydrodynamic grids were differentiated and classified as being on land or at sea, and the minimum distance between all the node coordinates and the shoreline data was calculated using the `geopy.distance` Python module (Geopy, 2022). For land nodes, distances were assigned negative values to ensure correct interpolation when crossing the land-

water boundary. Distance data for each node to the shoreline in each domain were added as netCDF4 files in the simulations, enabling the nesting of these files in the same manner as the nested hydrodynamic grids. The pre-processing required to prepare the distance grids is included in a series of Python scripts in the `scripts` folder.

A total of five particles were released every hour from each point for the period 1 March 2017 00:00:00 to 31 March 2017 23:00:00, with a total of 3,720 particles released from each point. In the first part of the sensitivity analysis with the two river

release points, a combined total of 7,440 particles were released in the domain. In the second part, where particles were released homogeneously there were a total of 491,040 particles released from 132 points (Table 3).





**Table 3.** Parameters, release points and total particles released for beaching sensitivity tests.

| Sensitivity Test | Beaching Scenario | Release points | Total Particles | Scenario parameters. |
|:---:|:---:|:---:|:---:|:---|
| 1 | 1 | River mouths | 7,440 | Velocity $\approx 0$ |
| 2 | 2 | River mouths | 7,440 | Distance $\leq 0$ |
| 3 | 3 | River mouths | 7,440 | Distance $< 1.694$ km, 6 h |
| 4 | 1 | Homogeneous | 491,040 | Velocity $\approx 0$ |
| 5 | 2 | Homogeneous | 491,040 | Distance $\leq 0$ |
| 6 | 3 | Homogeneous | 491,040 | Distance $< 1.694$ km, 6h |

## 3 Results

In this section, the main numerical experiments that were conducted will be presented. Firstly, the validation results of the
LOCATE model are presented, followed by a comprehensive beaching sensitivity test using different criteria with which to
define beaching parameters. Finally, beaching amounts and residence times of simulations using the same debris outflow data
are compared using nested grids and the IBI-CMEMS grid.

### 3.1 Lagrangian validation of LOCATE

The model validation process involved conducting simulations along the trajectory of the drifter data to qualitatively and
quantitatively compare the observed (L) and the simulated (D) trajectory distance and SS calculations using different prediction
horizons as illustrated in Fig.6. The simulation times had decreasing horizons of 1 hour with every time-step along the observed
trajectory, with all simulations having the same end-time. Results from the progression of the SS values at different horizons
are nuanced when comparing simulations conducted with nested grids and the IBI-CMEMS grid. For a 6-hour horizon, there
are two notable peaks of SS values, the first one at 13 hours (IBI-CMEMS SS=0.41) and 15 hours (nested grid SS=0.40)
from the initial simulation, and the second one at 43 hours for both (IBI-CMEMS grid SS=0.43, nested grid SS=0.53). Results
on a 24-hour horizon, demonstrate a better performance for nested grids with a peak at SS=0.49 at 13 hours from the initial
simulation, whereas the peak at 40 hours was higher with the IBI-CMEMS simulation, SS=0.69 compared to SS=0.52 for
the nested grid simulation. Overall, The 72-hour horizon was more stable with the nested grids with values exceeding 0.60
for simulations between 9 hours and 39 hours from the initial simulation, with a peak of 0.68 for the simulation at 13 hours.
Simulations with IBI-CMEMS displayed much greater variability, with a peak of SS=0.80 at 36 to 39 hours from the initial
simulation. The IBI-CMEMS simulation had lower SS values than the nested grid simulations for the first 27 simulations (out
of 48 total for each).





At a 72-hour horizon, the cumulative observed distance of the drifter (L) consistently exceeds the simulated cumulative distance (D) in both types of simulations, reaching a peak value of 1,805 km for L. In comparison, the highest D values were

792 km for the nested simulations and 1,264 km for the IBI-CMEMS simulation. Qualitatively, as depicted in Fig.6b the simulations using nested grids demonstrate a similar growth pattern, whereas the IBI-CMEMS simulations display periods of cumulative distance increases and decreases. The SS of the mean trajectory of the initial simulation (Fig.6c and Fig.6h) reflects the variability of the simulated trajectories. Notably, some beaching observed in plots Fig.6i and Fig.6j around the Barcelona city area, potentially indicative of the lower SS value in Fig.6e at 48 hours from the initial simulation. The SS values at 72

hours and 96 are very high (SS=0.83) with increasing cohesiveness between the simulated and observed trajectories as the simulations progress over the track of the drifter trajectory. It is important to note that these results are not reflected in the SS horizon plot (Fig.6a) or the cumulative distance plot (Fig.6b), as the 72-hour horizon was not available beyond the 48th simulation.



**Figure 6.** Lagrangian validation of LOCATE using data from drifter 6592 between 9 March 2022 and 14 March (maximum of 120 h) using the skill score test. Plot (a) illustrates the SS of the simulations at 6 h, 24 h and 72 h horizons using nested grids and the IBI-CMEMS grid. Plot (b) shows the time evolution of the cumulative distance (D) between the simulated and observed trajectories, and the observed cumulative distance (L) for the same simulation using nested grids and the IBI-CMEMS grid at 6 h, 24 h and 72 h horizons using a logarithmic scale. Plots (c) to (g) show the SS of mean trajectories for simulations with nested grids at 24 h intervals with their final SS, with a SS colour band for scale. Plots (f) to (j) show the trajectories of the simulated particles using nested grids as well as the drifter trajectory for comparison.

## 3.2 Beaching sensitivity analysis

Figure 7 illustrates the distance from the shoreline at which particles become beached according to the different criteria for each of the scenarios. As expected, tests with scenario 2 show particles closely aligning with the actual shoreline, while tests using scenario 1 follow the boundaries of the hydrodynamic grids of varying resolutions with current velocity data. The results presented in Table 4 indicate that the homogeneous particle release, serving as a control, yielded much lower beaching amounts (<21.06%) compared to releases from the two river release points close to the shore (86.48% to 94.88%). The highest beaching

amounts of each group were found in tests using scenario 3. The residence times for the tests using scenario 1 were notably



longer than for tests using scenario 2. Additionally, trajectory distances were greater in scenario 3 when particles were released close to the shoreline, although this trend was reversed in the homogeneous particle release. The value of 6.08 h for test 3 includes the 6-hour parameter for that scenario and the first time-step thereafter, set to 5 minutes.

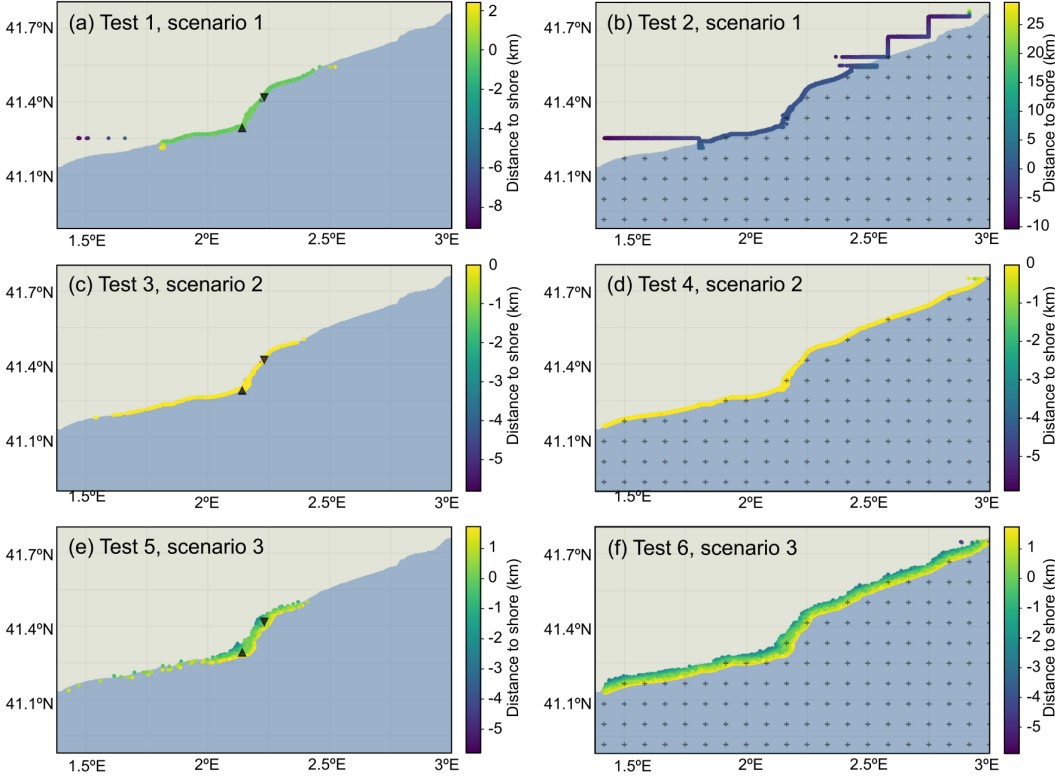

**Figure 7.** Locations of beached particles for beaching sensitivity tests. Markers represent particle release points, with a triangle up marker representing the Llobregat River, a triangle down marker representing the Besòs River, and the plus marker representing nodes on the IBI-CMEMS domain for the homogeneous particle release tests.





**Table 4.** Beaching amount percentage, median particle residence time and median particle trajectory with standard errors (SEM).

| Sensitivity Test | Beaching Scenario | Beaching amount (%) | Residence time (h) | Residence time SEM | Trajectory (km) | Trajectory SEM |
|---|---|---|---|---|---|---|
| 1 | 1 | 86.48 | 3.25 | 0.79 | 3.79 | 0.98 |
| 2 | 2 | 91.65 | 1.00 | 0.65 | 1.20 | 0.80 |
| 3 | 3 | 94.88 | 6.08 | 0.43 | 7.24 | 0.53 |
| 4 | 1 | 15.59 | 141.50 | 0.44 | 174.38 | 0.55 |
| 5 | 2 | 19.31 | 89.67 | 0.39 | 111.56 | 0.48 |
| 6 | 3 | 21.06 | 78.25 | 0.35 | 97.54 | 0.43 |

The snapshots presented in Fig.8 correspond to the simulation using nested grids, providing a visual representation of how the model uses hydrodynamic data of varying resolutions. For illustrative purposes, the month of May was selected corresponding to the period of highest particle release from the Llobregat River, as shown in Fig.4. The snapshots capture the dispersion of particles over 6-day intervals, starting from the beginning of the month. As the month progresses, the snapshots reveal the particles moving towards the coastline and dispersing along the majority of the coastal region within the domain. Some particle

accumulation was observed towards the south of the domain on 31 May 2017 due to boundary conditions where the mean current velocity for that day was close to 0 m s$^{-1}$ in the nodes west of 1.6°E between 41.2°N and 40.95°N. The particle density maps highlight the highest densities observed close to the release points, as evidenced on 7 May 2017 and 31 May 2017.





**Figure 8.** Simulation snapshots using nested grids for dates in the month of May 2017 at 23:00 hours. Maps on the left show the dispersion of particles released. Maps on the right show particle density of the number of particles km$^{-2}$.

As can be inferred in Fig.9, the total beaching amounts observed in both simulations were very high. The nested grid simulation revealed a beaching proportion of 91.5%, with 8.5% of particles being exported, while the simulation employing



the IBI-CMEMS grid exhibited a beaching proportion of 95.8%, with 4.2% of particles being exported. When examining the beached quantities within the demarcated zones in Fig.9a, there were marked differences between both simulations. The Prat de Llobregat area had 12.7% more particles from the IBI-CMEMS simulation, whereas the Llobregat River mouth showed 8.8% more beached particles with the nested grid simulation.

    The larger beaching differences were found in regions of complex shoreline configuration. For instance, the amount of
beaching was over 10 times higher in the external port area with the nested grid simulation, and 2.5 times higher inside the port area. Barceloneta Beach experienced over 6 times as many beached particles with the nested grid simulations, and the other city beaches exhibited over 2.5 times as many beached particles. The number of beached particles in these areas, however, was lower than in other areas such as the Llobregat River mouth and surrounding areas. Residence times increased substantially as the zones moved further away from the release points, particularly evident in the areas south of Castelldefels
and north of Montgat. The increase in residence times was also observed between release points where the higher resolution grids provided the hydrodynamic data. Particle residence times consistently showed higher values for the simulation using the nested grids, with the inside of the port area registering 18 times higher values when compared to the same area in the IBI-CMEMS simulation.

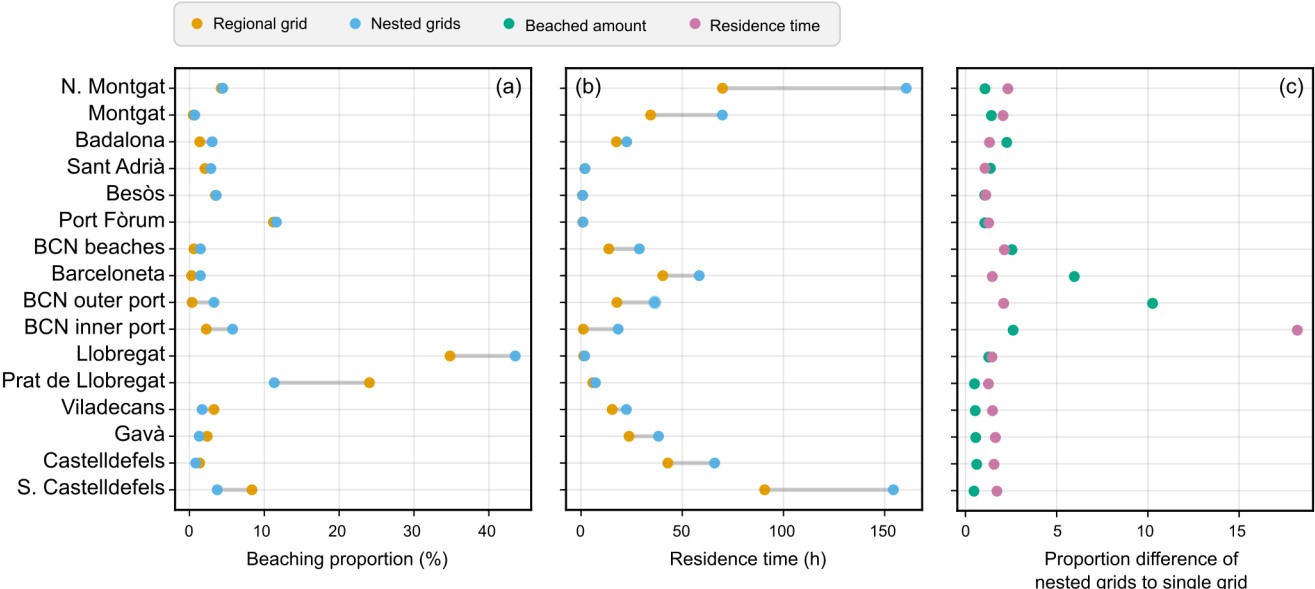

**Figure 9.** (a) Beaching amounts per demarcated zone as a percentage of the total amount of beached particles for the nested grids and IBI-CMEMS grid simulations. (b) Residence time median per demarcated zone for the nested grids and the IBI-CMEMS grid simulations. (c) Proportion difference of beaching amount and residence time between simulations using nested grids and the IBI-CMEMS grid.





## 4 Discussion

### 4.1 Numerical models and coastal processes

Numerical models of marine debris transport to date have been valuable in providing debris budget estimates, measuring fluxes and studying interconnectivity at oceanic scales (Chassignet et al., 2021; Eriksen et al., 2014; Law et al., 2010; Lebreton et al., 2012; Maximenko et al., 2012; Onink et al., 2019, 2021; van Sebille et al., 2012, 2015). Some studies have focused on semi-enclosed basins such as the Mediterranean Sea to establish debris dispersion and accumulation patterns (Kaandorp et al., 2020; Liubartseva et al., 2018; Mansui et al., 2015, 2020; Zambianchi et al., 2017). The models used in the aforementioned studies use similar coarse hydrodynamic resolutions, typically of $\geq 2.5$ km. The application of Lagrangian models to nearshore systems with complicated geometries and at smaller scales is much less mature than oceanic scale models. This is likely due to the complexities involved and that modelling of particle beaching in a shoreline defined by coarse resolutions can be difficult, leading to inconsistencies or inaccuracies (Critchell et al., 2015; Critchell and Lambrechts, 2016; Neumann et al., 2014; Yoon et al., 2010; Zhang, 2017). To address these limitations, nesting hydrodynamic grids with varying resolutions enables more accurate simulations in high-interest areas such as harbourss and adjacent coastal waters, where higher-resolution data may be available. Even if the high-resolution data does not encompass the entire study domain, large parts of the study domain can be covered. In the present study, the coastal grid spans all the demarcated areas to at least some extent as illustrated in Fig.3.

The results from the Lagrangian validation of the model indicate that the accuracy of the simulated trajectories using LOCATE is highly dependent on the resolution of the Eulerian hydrodynamic currents data. The model's validation results are consistent with a study by Castro-Rosero et al. (2023) in the Black Sea which also demonstrated the suitability of LOCATE for predicting the motion of floating marine debris. Furthermore, LOCATE exhibits high skill stability and performance for the 72-hour horizon utilising nested grids, a commonly used threshold in other studies to determine the sensitivity of the SS test (Révelard et al., 2021). The comparison of simulations conducted using nested grids and the IBI-CMEMS grid across all horizons presented somewhat mixed results, although SS values for the 72-hour horizon were higher for 56.25% of the simulations conducted. This is indicative of the challenges associated with predicting trajectories close to the shoreline influenced by coastal processes, amplified by the strong influence of the alongshore northern current as seen south of the Barcelona city area in Fig.6h to Fig.6l. While the performance of the model can be effectively validated due to good SS values using nested grids, drawing direct comparisons between nested grids and the IBI-CMEMS grid using the SS test is challenging. Within a single trajectory, the influence and contribution of each grid as a particle moves across different domains cannot be isolated even if it is possible to numerically determine which grid has provided the hydrodynamic data for that time-step, due to the cumulative nature of the test. Additionally, an area of future work would be to address the paucity of available and suitable drifter information at coastal scales where high-resolution hydrodynamic data may apply, to increase the statistical robustness of additional validation analyses.

In the simulation comparison between nested grids and the IBI-CMEMS grid, the particle release coordinates coincided with coastal nodes close to the midpoint of the Llobregat and Besòs River mouths to prevent excessive accumulation of particles at the river mouths from possible lack of hydrodynamic data at those specific points. While the coastal grid data includes





river contributions in the form of climatological run-off values and a constant salinity field, the actual outflow rates at the river mouths (mean values of 20.77 m$^3$ s$^{-1}$ for the Llobregat River, and 4.33 m$^3$ s$^{-1}$ for the Besòs River) were not included.

Indeed, the real-time inclusion of discharge observations or inputs from hydrological forecast models is currently undergoing development in the SAMOA system. Once implemented, this enhancement would allow forcing to an extended run-off rate at every coastal grid point, thereby increasing the accuracy of simulations in areas adjacent to the river mouths where very high beaching rates can be found (Sotillo et al., 2020).

Numerical models that only use low-resolution hydrodynamic grids have been shown to inadequately reproduce subme-
soscale structures, particularly in the Western Mediterranean region, causing model underperformance and potential problems when tracking maritime emergencies (Sotillo et al., 2021). Aside from the inner port area, as depicted in Fig.9c there are elevated residence times and beaching amounts in all other areas where the harbour grid applies which also exhibit complex coastal structures such as beaches and groynes, such as the external port area, Barceloneta beach, and other Barcelona beaches. Higher residence times are also observed in other areas particularly those close to the study domain limits, likely due to the
contribution of longer residence times of particles when traversing the higher-resolution domains during the simulation. The beaching differences in Fig. 9a also illustrate variability in some areas. For instance, the Llobregat River mouth shows 8.7% more beaching when using nested grids (43.5% compared to 34.8%), whereas the Prat de Llobregat, an adjacent area to the south of the river mouth, experienced over twice the amount of beaching (24.0% compared to 11.3%) when only using the IBI-CMEMS grid. This observation suggests that the IBI-CMEMS grid may have higher current velocities around the Llobregat
River mouth with respect to the higher-resolution grids, leading to the transport of more particles to the neighbouring area to the south.

The present study yields findings consistent with those reported by (Sotillo et al., 2021), where residence times are substantially increased up to 18 times when high-resolution data are applied in areas of topographic complexity such as the inner port area, even though residence times are relatively low. By incorporating high-resolution data and utilizing a detailed coastline to
parameterise particle beaching, it becomes possible to distinguish these complex structures, even in regions with high fractal dimensionality and natural barriers. Conversely, lower spatial resolutions may lead to an underestimation of beaching amounts in areas containing complex structures or adjacent to them. The external port area, which is separated by a barrier from the internal port area, exhibits markedly higher beaching amounts when high-resolution grids apply. Thus, a distinctive feature of the Barcelona harbour which has two mouths separated by a quay increasing the complexity of the hydrodynamic behaviour
of the port is considered when applying the harbour grid (Sotillo et al., 2021). Solely relying on large-scale, low-resolution grids such as IBI-CMEMS is insufficient for coastal-scale simulations, although some limitations exist in the high-resolution hydrodynamic data utilised in this study.

The hydrodynamic data used in this study does not include wave-induced currents, therefore excluding some important coastal processes such as longshore currents that can be very important in the transport, deposition and redeposition of material
in areas potentially far away from the point of emission. Despite the highest resolution grid used resolving down to 70 m, it is still not sufficient to resolve undertow or rip currents and resulting gyres that not only shape sandy shorelines but also contribute to the transport of material offshore especially where convergence occurs by longshore currents (Hinata and Kataoka, 2016).



Currently, the wave element of the hydrodynamic data is a one-way coupled system between the regional CMEMS physical model (IBI-PHY) and the IBI wave products (IBI-WAV) that includes the Stokes drift, wave-induced mixing and wind drag coefficient formulas based on sea states. This wave data, however, is not included in the hydrodynamic data provided by the SAMOA application although efforts are being made to incorporate this (García-León et al., 2022). Indeed, a positive impact has been observed in model predictions from coupling wave-current data in open water and shallower coastal areas especially during extreme weather events such as Storm Gloria in January 2020 (Sotillo et al., 2021). In the present work, the Stokes drift component is provided by the IBI-WAV system, which has an even lower resolution (1/20°) than the IBI-CMEMS grid (1/36°) and is not integrated into the nested solution (CMEMS, 2023).

The use of LOCATE to perform Lagrangian simulations using nested hydrodynamic grids is not limited to the current study domain and could be transposed to other areas where hydrodynamic data in varying resolutions may be available. Numerical simulations of other coastal regions in Spain are hosted on the PdE website making the extension of the developed system to other regions relatively simple (García-León et al., 2022; Sotillo et al., 2015, 2021). Hydrodynamic grids may still need to be adapted to work in conjunction with each other as shown to ensure that grid structure, orientation and time data are compatible.

Currently, plastic dispersion simulations assume that the driving force behind the movement of plastic particles is a linear combination of Stokes drift and mean currents. While this approach is realistic in mid-low energetic conditions and in coastal regions relatively far from the coastline (100 m from the coastline to the continental shelf), it fails to account for wave-induced currents and processes that are not resolved by current models, having important implications regarding the movement and transport of material. Moreover, the spatial resolution constraints result in particles moving linearly towards land without being caught up in the surf and swash areas, potentially leading to underestimations of residency times (Hinata et al., 2017).

Another drawback of not including wave processes is that particle resuspension and subsequent redeposition events cannot be effectively parameterised in a deterministic manner given the scarcity of data to form the basis of probabilistic definitions. To address these limitations, a beach domain based on the Coupled-Ocean-Atmosphere-Wave-Sediment Transport Modelling System (COAWST) numerical model with a resolution of 23 m is being developed at the Universitat Politècnica de Catalunya. This grid will be nested into the harbour domain enabling very detailed simulations that can accurately characterise coastal processes. Wave-current interaction processes will be relevant in this domain as the transport of plastic particles in coastal environments is largely influenced by wave-induced motions, with waves in coastal regions being highly non-linear as a consequence of wave-topography interaction.

## 4.2 Particle beaching

The effect of wind on the currents is already incorporated into the coastal and harbour grids which have atmospheric forcing by AEMET and can have a significant influence on beaching amounts, with small changes even in the short term causing important effects on the trajectories and accumulation patterns (Critchell et al., 2015; Rosas et al., 2021). Wind drag on particles, however, is not considered since it would require particle size, shape and buoyancy to be determined for virtual particles which are assumed to be floating just beneath the surface. The high beaching values in both simulations are higher than other recent studies which are dependent on region, scale, and simulation period. As can be seen in Appendix A, the beaching amounts





are highly variable with some studies from the central and eastern Mediterranean Sea such as Politikos et al. (2017, 2020) recording very low beaching rates, while other studies such as Macias et al. (2019) observed amounts higher than the present study. Other prominent studies at oceanic scales such as Onink et al. (2021) and Lebreton et al. (2019) recorded lower beaching

amounts than this study, with 77% and 67% respectively, over integration times that span up to 20 years. Unfortunately, there is not enough comparable beaching data from other comparable numerical modelling studies at similar coastal scales as of yet.

Defining the parameterisation and inclusion of particle beaching in numerical models is crucial for identifying areas within the coastal zone that may experience increased pressure from plastic pollution and the range of these different types of definitions can be seen in Appendix A. It is important to note that definitions used in larger-scale studies may not be suitable for

localised analyses. Deterministic parameterisations, which are simpler in formulation than probabilistic parameterisations, often depend on whether a particle crosses the land-water boundary, such as Lebreton et al. (2012), Macias et al. (2019), Macias et al. (2022), Politikos et al. (2017, 2020), Rosas et al. (2021), and Ruiz et al. (2022), and are represented under scenario 1. Another common deterministic parameterisation involves a predetermined distance travelled over a certain amount of time to indicate particle stagnation, such as Chassignet et al. (2021), Lebreton et al. (2019), and Mansui et al. (2015), and are rep-

resented under scenario 3. The beaching sensitivity analysis demonstrates the difficulty of detecting complex geometries at coastal scales when using current velocity, as seen in the tests that use scenario 1 (Fig.7a and Fig.7b). In this scenario, shoreline identification had different resolutions along the coastline, resulting in an overestimation of residency times and trajectory distances with respect to using a physical land-water boundary such as under scenario 2. While higher resolution grids using current velocity in scenario 1 can adequately resolve areas like the inner port, the lower resolution grids can cause uncertainty,

with particles potentially travelling several kilometres inland before being considered beached. This uncertainty hinders the determination of areas most at risk of receiving and accumulating marine debris. Furthermore, scenario 3, which utilises a time and distance requirement, introduces additional uncertainty regarding the likelihood of beaching occurrence, with higher residence times and trajectory distances also observed. Additionally, particles drifting alongshore within the specified distance would be immediately considered beached after the specified time, effectively disregarding hydrodynamic conditions or data

that may solve coastal processes (if these are included) within the distance parameter, which could explain the higher beaching values observed. Conversely under this scenario, if a particle reaches a land cell where there is no velocity data before the time limit, it would continue to move due to the horizontal diffusivity that continues the stochastic movement of a particle, resulting in overestimations of residence times.

Scenario 2 introduces a different deterministic beaching model that relies on a physical shoreline and calculates the distance

between virtual particles and the shoreline to determine when they cross the land-water boundary as illustrated in the beaching sensitivity analysis (Fig.7c and Fig.7d). This parameterisation enables accurate and precise detection of particle beaching along the shoreline at smaller coastal scales, providing a distinct approach compared to other methods for detecting land or particle stagnation. Using the distance to the shoreline and a physical boundary detection ensures consistency throughout the study domain, even when nesting grids of varying resolutions. Additionally, the interpolation capabilities of Parcels allow

real-time calculation of the minimum distance to the shoreline and are not limited by a decrease in hydrodynamic spatial resolution. Shoreline detection based on a physical boundary does have some limitations, such as relying on the availability of





high-resolution spatial data and requiring preprocessing steps. While obtaining detailed coastline data may not be feasible for larger-scale studies, the advantages it offers for smaller-scale studies are substantial, particularly when assessing the impact of beaching at a localised level.

Across all scenarios, there were noticeable differences in beaching, residence times and trajectory distances between the tests that involved two release points from the river mouths and the homogeneous release used as a control mechanism. As expected, the beaching amounts were far higher from the river release points due to their proximity to the shore. This outcome confirms that coastal processes are incorporated in the simulations and that these favour the shoreward movement of particles, especially when they are already close to shore.

## 565   5   Conclusions

The LOCATE model focuses on simulating the dispersion of floating marine debris in coastal waters and the continental shelf region near the coastline. It uses high-resolution hydrodynamic data to allow the study of debris exchange from coastal to offshore regions resolving coastal processes within Lagrangian simulations. In the present work Lagrangian simulations have been conducted using hydrodynamic data from IBI-PHY and IBI-WAV products from CMEMS, as well as high-resolution

data from the SAMOA application. The LOCATE model effectively integrates high-resolution hydrodynamic data within lower-resolution grids around high-interest areas without substantial or perceptible computational cost.

A Lagrangian validation of the model by comparing simulated trajectories with an observed trajectory of a CODE drifter demonstrated good skill and confidence in consistently predicting the movement of debris when using nested grids, including a better prediction of the distribution of particles when there is a complex topography. Moreover, the use of high-resolution

hydrodynamic data enabled the model to resolve complex geometrical structures that would otherwise be indistinguishable using coarser resolution data, thus providing greater confidence in estimating residence times and trajectory distances. The nesting of hydrodynamic grids of varying resolutions also overcomes problems with particles quickly moving outside of the high-resolution grid boundaries by seamlessly integrating these grids.

Currently, there are still some limitations in the hydrodynamic data available that do not allow for all coastal processes to

be included, particularly those derived from wave action or particle behaviours in the surf and swash zones. These behaviours, however, could be incorporated into the present model once they are available in the hydrodynamic data. It has been demonstrated that using real-time particle distance to the shoreline and a physical land-water boundary can accurately model particle arrival time and beaching locations, which becomes more salient the smaller the scale of the study. The beaching process of marine debris is still not fully understood, and the inclusion of resuspension and redeposition events would require either

higher resolution data capable of resolving wave-induced coastal processes or a more probabilistic approach despite a paucity of information on resuspension timescales. Despite these limitations, the LOCATE model provides for accurate depictions of accumulation zones and debris hot-spots in localised settings which could be transposed to other areas where high-resolution data may be available.



*Code availability.* The code for LOCATE is archived at Zenodo at https://doi.org/10.5281/zenodo.8345027 (Hernandez et al., 2023). The
code is also available through a GitHub repository found at https://github.com/UPC-LOCATE/LOCATE/.





## Appendix A: Beaching parameterisation

**Table A1.** Marine debris particle beaching parameterisations and amounts in the literature. P = probabilistic, D = deterministic

| Author | Metocean data | Reso-lution | Beaching type | Parameterisation | Integration time | Beaching amount |
|---|---|---|---|---|---|---|
| Chassignet et al. (2021) | HYCOM NCODA | 1/4° | D | Distance condition over 30 days | 10 y | 75.4% |
| Kaandorp et al. (2020) | CMEMS | 1/16° | P | Probability that a particle spends time in a shore-adjacent cell. | 10 y | - |
| Kaandorp et al. (2022) | CMEMS | 1/9° | P | Probability that a particle spends time in a shore-adjacent cell. | 5 y | - |
| Lebreton et al. (2012) | HYCOM NCODA | 1/12° | D | Drifted into a land cell | 30 y | <40.0% |
| Lebreton et al. (2019) | - | 1/16° | D | 2 days < 1/16°(6.4 km) | 20 y | 66.8% |
| Liubartseva et al. (2018) | NEMO WW3 CMEMS | 1/16° | P | Drifted to a land cell, but has a 'washing off' probability algorithm for resuspension. Considers 5 resuspension events before beaching. | 4.5 y | - |
| Macias et al. (2019) | GETM | 9 km | D | Drifted into a land cell | 10 y | 98.7%-99.8% |
| Macias et al. (2022) | GETM | 9 km | D | Drifted into a land cell | 17 y | - |
| Mansui et al. (2015) | NEMO | 1/12° | D | Beached if velocity over 14 days ≤ 0.5 cm s$^{-1}$ | 10 y | - |
| Onink et al. (2021) | HYCOM NCODA | 1/12° | P | Beaching probability based on beaching timescale of 63.2% within 10 km of shore. Considers resuspension. | 10 y | 77.0% |
| Politikos et al. (2017) | POSEIDON POM | 1/15° | D | Drifted into a land cell | 1 y | 5.6%-13.8% |
| Politikos et al. (2020) | POSEIDON POM | 1/20° | D | Drifted into a land cell | 3 y | 9.2% |
| Rosas et al. (2021) | SOMA | 1 km | D | Drifted into a band 5 m from a land cell | 10 m | 78.0% |
| Ruiz et al. (2022) | TESEO | 0.08° | D | Drifted into a land cell | 1 y | 80.0% |
| Yoon et al. (2010) | OOPS | 1/12° | D | - | 4 y | - |
| Zambianchi et al. (2017) | - | 1/2° | - | - | 6 y | 59.0% |





## Appendix B: Regional node coordinates

**Table B1.** Regional node coordinates used for the release of particles in the beaching sensitivity test.

| Latitude (°N) | Longitude (°E) |
|---|---|
| 40.91666794 | 1.41666579, 1.49999917, 1.58333254, 1.66666579, 1.74999917, 1.83333254, 1.91666579, 1.99999917, 2.08333254, 2.16666579, 2.24999905, 2.33333254, 2.41666579, 2.49999905, 2.58333254, 2.66666579, 2.74999905, 2.83333254, 2.91666579, 2.99999905 |
| 41 | 1.41666579, 1.49999917, 1.58333254, 1.66666579, 1.74999917, 1.83333254, 1.91666579, 1.99999917, 2.08333254, 2.16666579, 2.24999905, 2.33333254, 2.41666579, 2.49999905, 2.58333254, 2.66666579, 2.74999905, 2.83333254, 2.91666579, 2.99999905 |
| 41.08333206 | 1.41666579, 1.49999917, 1.58333254, 1.66666579, 1.74999917, 1.83333254, 1.91666579, 1.99999917, 2.08333254, 2.16666579, 2.24999905, 2.33333254, 2.41666579, 2.49999905, 2.58333254, 2.66666579, 2.74999905, 2.83333254, 2.91666579, 2.99999905 |
| 41.16666794 | 1.49999917, 1.58333254, 1.66666579, 1.74999917, 1.83333254, 1.91666579, 1.99999917, 2.08333254, 2.16666579, 2.24999905, 2.33333254, 2.41666579, 2.49999905, 2.58333254, 2.66666579, 2.74999905, 2.83333254, 2.91666579, 2.99999905 |
| 41.25 | 1.91666579, 1.99999917, 2.08333254, 2.16666579, 2.24999905, 2.33333254, 2.41666579, 2.49999905, 2.58333254, 2.66666579, 2.74999905, 2.83333254, 2.91666579, 2.99999905 |
| 41.33333206 | 2.16666579, 2.24999905, 2.33333254, 2.41666579, 2.49999905, 2.58333254, 2.66666579, 2.74999905, 2.83333254, 2.91666579, 2.99999905 |
| 41.41666794 | 2.24999905, 2.33333254, 2.41666579, 2.49999905, 2.58333254, 2.66666579, 2.74999905, 2.83333254, 2.91666579, 2.99999905 |
| 41.5 | 2.41666579, 2.49999905, 2.58333254, 2.66666579, 2.74999905, 2.83333254, 2.91666579, 2.99999905 |
| 41.58333206 | 2.58333254, 2.66666579, 2.74999905, 2.83333254, 2.91666579, 2.99999905 |
| 41.66666794 | 2.83333254, 2.91666579, 2.99999905 |
| 41.75 | 2.99999905 |



*Author contributions.* Author contributions: Ivan Hernandez: Data curation, formal analysis, investigation, methodology, programming, visualisation. Writing - original draft. Leidy M. Castro-Rosero: Methodology, validation, analysis and programming. Writing - review and editing. Manuel Espino: Conceptualisation, methodology, supervision. Writing - review and editing. Jose M. Alsina Torrent: Conceptualisation, funding acquisition, methodology, model development, visualisation, supervision. Writing - review and editing.

*Competing interests.* The authors have no conflicts of interest to declare. All co-authors have seen and agree with the contents of the manuscript and there is no financial interest to report. We certify that the submission is original work and is not under review at any other publication.

*Acknowledgements.* The present study was developed within the TRACE (Tools for a better management of marine litter in coastal environments to accelerate the tRAnsition to a Circular plastic Economy) project (TED2021-130515B-I00) funded by the Spanish Science and Innovation Ministry (MCIN/AEI/ 10.13039/ 501100011033) and by the EU "NextGenerationEU"/PRTR. The development of the presented model was initiated within the LOCATE (Prediction of plastic hot-spots in coastal regions using satellite derived plastic detection, cleaning data and numerical simulations in a coupled system) project funded by the European Space Agency ESA within the Open Space Innovation Platform (OSIP) campaign (Contract No. 4000131084/20/NL/GLC). IH acknowledges funding from Formació de Professorat Universitari (FPU-UPC 2020). LCR acknowledges funding from Ministerio de Ciencia y Tecnología - Scholarship Program No. 885. JA acknowledges funding from the Serra Húnter Programme (SHP).



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
