# Peer review of "LOCATE v1.0: Numerical Modelling of Floating Marine Debris Dispersion in Coastal Regions Using Parcels v2.4.2"

_Geoscientific Model Development, 2023_

## Referee Comment (RC1)

**Hernandez *et al.* Review**

**General comments**

Hernandez *et al.* introduce a modelling framework (LOCATE) for simulating marine dispersal in coastal regions using nested hydrodynamic grids. The authors apply LOCATE to a nested model of the Barcelona coastline (forced by a regional CMEMS model), and compare trajectory forecasting skill and beaching behaviour between the (higher resolution) nested model and (lower resolution) regional model, as well as the sensitivity of beaching to three beaching parameterisations.

Although LOCATE has been used in previous studies, this manuscript represents a considerable effort to document, describe, and illustrate a potential application for the framework. This is a great example of 'open science', and the quality of figures is also very high. Unfortunately, in its current form, the manuscript suffers from a lack of purpose and clarity, and feels more like a detailed supplement to Castro-Rosero *et al.* (2023). If the main purpose of the manuscript is to facilitate the wider adoption of LOCATE, the manuscript should clearly explain the advantages of LOCATE over just using Parcels directly, since the nested grid capability of Parcels is already quite accessible (i.e. if I were to investigate dispersal across nested grids, it is not clear to me why I would use LOCATE rather than 'pure' Parcels). If the main purpose is to instead explore the utility of nested grids for particle tracking in coastal waters, more thorough validation is needed than comparison with a single drifter profile. I realise that there is limited observational data available given the small domain size (using drogued drifters may help), but many of the manuscript's claims about "model accuracy" are weakly supported by evidence, and the manuscript therefore provides limited insights into the (dis)advantages of using nested grids beyond "there are differences". I also have concerns about some aspects of the methods, and other aspects were very difficult for me to understand.

If the authors can address the general comments above, then I would recommend major revisions to the manuscript (please see the attached document for specific and technical comments).

**Specific comments**

- o The abstract is long compared to the number of key messages from the paper. I would recommend condensing the abstract to make it easier for readers to identify the main points.
- o Section 1.2 covers the dynamics of beaching debris in great detail, but most is of little relevance to the paper (since most of these dynamics are not investigated). I would recommend condensing.
- o Section 1.4 begins by stating that the "goal of the present work is to simulate particles taking into account coastal processes using nested hydrodynamic grids at varying resolutions". I would argue that this is the *method*, not the *goal*. As stated in the general comments, I would recommend clarifying what the actual goal of the paper is.
- o Section 2.2: What is the frequency of the Eulerian model output used in LOCATE? Line 159 states that "data are calculated daily" – daily strikes me as very low compared to the timescales over which currents resolved by a 2.5 km model (and certainly 350/70 m) change.
- o I cannot see any reference to tides in the manuscript outside of the introduction. Were tides not included in these simulations and, if not, why?
- o Section 2.3: Why was model output interpolated to a 'regular' grid (presumably A-grid)? As described in the OceanParcels documentation[1], A-grids introduce interpolation problems whereby particles get artificially stuck at the model coastline, because both velocity
* * *
[1] https://docs.oceanparcels.org/en/latest/examples/documentation_stuck_particles.html

components approach zero at the coastline (i.e. particles will always stagnate at the coastline of an A-grid, regardless of the strength of any along-shore currents). Particularly given the focus of this manuscript on beaching, I am concerned about the potential for beaching to occur due to interpolation artefacts, rather than any physical reason. The authors could consider using the original C-grids instead, or the free-slip boundary condition that was recently implemented in Parcels[2].

o   Section 2.5: Given that LOCATE uses a constant value for $K$, most of the detail in lines 233-243 is unnecessary and should be removed. I am also confused by the choice of a constant value for $K$ (10 m$^2$ s$^{-1}$). Should this not be scale-dependent? Okubo (1971) would suggest that a value of 10 m$^2$ s$^{-1}$ would be appropriate to reproduce the effects of unresolved scales of motion at a resolution of around 10 km, much coarser than the resolution used in this study.

o   Section 2.7: I am surprised by the decision to use undrogued drifters. Drifters lacking a drogue experience a significant direct push from the wind due to the exposed part of the drifter (Poulain et al., 2009), which is not included in LOCATE. Although the authors' concern about velocity shear within the upper water column is valid, there is some evidence that surface currents reasonably represent the forces driving drogued drifter movement (Imzilen et al., 2023). Using drogued drifters may also increase the number of drifters available to the authors for use in validation.

o   Section 2.7: I am uncomfortable with the use of the word "validation" for the comparison with a single drifter. Is this really enough data to count as validation?

o   Section 2.7: How were drifter trajectories temporally interpolated if there are no timestamps associated with locations?

o   Section 2.8: I cannot find a precise definition for the residence time in this manuscript – please clarify how this was calculated.

o   Section 2.8: Since the manuscript does not investigate or validate the temporal variability of debris accumulation, I do not see the point of varying the input rate of debris.

o   Section 2.9: If I understand correctly that scenario 2 defines the distance-to-shore relative to the true (rather than model) coastline, would this not make it impossible for debris to beach in many places? There are presumably parts of the model grid where ocean cells do not quite reach the true coastline (see below), so it is never possible for the distance-to-shore parameter to reach zero here, and therefore also impossible for particles to beach? In any case, I do not understand the point of using the real coastline in a beaching parameterisation – particles within the simulation do not 'know' anything about the real coastline, they only 'know' about the model coastline.

[Figure]

**True land**
**Model coast**
**Unbeachable coast**
* * *
[2]   https://docs.oceanparcels.org/en/latest/examples/documentation_unstuck_Agrid.html#3.-Slip-boundary-conditions

- Section 2.9: I do not understand scenario 3. Lines 533-535 makes it sound like particles beached if they travelled less than 1.694 km in 6 hours, but I am not sure if this is correct.
- Section 3.1: I did not understand this section (particularly what was meant by 'horizons') until I read Révelard et al., (2021). Please clarify (probably in the methods) the methods, i.e. comparing the skill of the model to *forecast* the trajectory of a drifter 6/24/72 h ahead, as a function of when the forecast started.
- Figure 6: How is the 'mean trajectory' defined? Is this just the arithmetic mean of all latitudes and longitudes at a point in time?
- Figure 7: Please use a divergent colourmap for these figures, centred at 0. The use of a sequential colourmap makes it very difficult to tell which particles are close to the coastline, versus which have over/undershot.
- Section 3.2: Please specify which beaching scenario was used for Figure 9 and lines 403-418.
- Figure 8: The left and right panels do not seem to correspond in this figure. For example, lots of blue particles appear in the far SW corner on May 31 (left), but not in the panel on the right.
- Section 4.1: Is it not surprising that the regional grid did not perform considerably worse than the nested grid? This is one of the most interesting observations in the manuscript for me (that the regional grid often had similar performance to the nested grid), and seems to somewhat contradict the manuscript's conclusion that using nested grid improves the accuracy of predictions (although there is insufficient validation in the manuscript to be sure).
- Section 4.2: Line 540 states that, under scenario 1, particles can travel "several kilometres inland before being considered beached". Along similar lines to my comment on Section 2.9, does this really matter? Is it really a problem that beaching locations are 'wrong' by a few kilometres in a model with a resolution of a few kilometres?
- Section 5: The manuscript does not validate any beaching predictions, so there is no data to support the claim that "using real-time particle distance to the shoreline… can accurately model particle arriving time and beaching locations". Similarly, no data has been presented to support the claim that "LOCATE… provides accurate depictions of accumulation zones and debris hot-spots…", at least not in this manuscript.
- Table A1: This table may not set out to be exhaustive but, in case of interest (as this is quite a nice compilation of beaching parameterisations that could be useful for others), some other studies using 'deterministic' parameterisations include Bosi et al., 2021; Cardoso & Caldeira, 2021; Critchell & Lambrechts, 2016; Dobler et al., 2019; Seo & Park, 2020; Zhang et al., 2020. Some other studies using 'probabilistic' parameterisations include Kaandorp et al., 2023; van der Mheen et al., 2020; Vogt-Vincent et al., 2023.
- Did the authors observe any artefacts in particle concentrations along the nested domain boundaries? There are quite striking discontinuities in surface velocity along domain boundaries in Figure 2, and it would be interesting to know whether this causes any artificial particle convergence/divergence at these boundaries.

**Technical comments/corrections**
- Lines 29-31: The number of references for the physics of marine dispersal is excessive in my view. I would recommend condensing, e.g. just using van Sebille et al., (2020).
- Lines 31-33: It is not clear to me why this sentence is relevant. Lines 31-36 could be removed for brevity.
- Line 40: This is a very specific range (50-600 m). Where is this range from?
- Line 42: Are density gradients really a *driver* of coastal currents (is the driver not the process that generated the density gradients in the first place, e.g. upwelling/downwelling)?
- Line 45: The referenced paper by Stokes was published in 1880, not 2009!

- Line 70: Would recommend changing "high spatial discretisation" to "fine spatial discretisation".
- Line 82: I do not follow how this relationship is exponential. Computational costs should scale broadly linearly with the area modelled, and a polynomial (not exponential) relationship with the model resolution.
- Lines 160-161: Please separate the references for ROMS, IBI-CMEMS, and SAMOA.
- Line 172: Typo ('downlaod')
- Line 174-178: I would recommend giving the resolution of the wave model here (I assume 1/20 degree, based on line 494).
- Line 198: It is not clear what is meant by "typical stochasticity". I assume the authors meant subgrid scale diffusion.
- Line 242-243: Citing OPeNDAP is unnecessary here (the text is describing the data itself, not OPeNDAP).
- Line 246: Please state or cite where the value (1/3) for the variance of the random process comes from.
- Table 3: It is a bit confusing using the same numbering for sensitivity test and beaching scenario. It may be clearer to remove the sensitivity test numbering, and change sensitivity test 1 to test 1R (beaching scenario 1, river release), test 2 to 2R, test 4 to 1H, etc.
- Figure 6 caption, last line: I assume "(f) to (j) should be "(h) to (l)".
- Lines 365-402: This paragraph is unrelated to beaching, so should not be in section 3.2.
- Line 404-405: This line implies that no particles were retained (at sea, unbeached). Is this correct?
- Line 406-408: Although it is possible to work this out from context, please specify that "The Prat de Llobregat area *had* 12.7% more particles…" means that 12.7% more particles *beached*. This sentence also makes it sound like these are relative percentages, whereas based on Figure 9 it looks like these are absolute percentages.
- Line 431: Typo ("harbourss")
- Lines 439-441: Please state for which configuration the SS value was higher (presumably the nested grid).
- Line 483: I would recommend replacing "wave-induced currents" with "wave-induced Eulerian (mean) currents". The Stokes drift is not a current, but some readers might be confused.
- Line 520: Please specify that "both simulations" refers to the regional vs nested grids.
- Lines 570-571: The manuscript does not discuss the difference in computational cost between the regional and nested configurations, so I would recommend removing this sentence (or quantifying the difference in computational cost, and moving this to the discussion).
- The labels "IBI-CMEMS" and "regional grid" are used interchangeably in this manuscript, which can get confusing. I would stick to one of them.

**References**

Bosi, S., Broström, G., & Roquet, F. (2021). The Role of Stokes Drift in the Dispersal of North Atlantic Surface Marine Debris. *Frontiers in Marine Science*, *8*(August), 1–15. https://doi.org/10.3389/fmars.2021.697430

Cardoso, C., & Caldeira, R. M. A. (2021). Modeling the Exposure of the Macaronesia Islands (NE Atlantic) to Marine Plastic Pollution. *Frontiers in Marine Science*, *8*(April). https://doi.org/10.3389/fmars.2021.653502

Critchell, K., & Lambrechts, J. (2016). Modelling accumulation of marine plastics in the coastal zone; what are the dominant physical processes? *Estuarine, Coastal and Shelf Science*, *171*, 111–122. https://doi.org/10.1016/j.ecss.2016.01.036

Dobler, D., Huck, T., Maes, C., Grima, N., Blanke, B., Martinez, E., & Ardhuin, F. (2019). Large impact of Stokes drift on the fate of surface floating debris in the South Indian Basin. *Marine Pollution Bulletin*, *148*(May), 202–209. https://doi.org/10.1016/j.marpolbul.2019.07.057

Imzilen, T., Kaplan, D. M., Barrier, N., & Lett, C. (2023). Simulations of drifting fish aggregating device (dFAD) trajectories in the Atlantic and Indian Oceans. *Fisheries Research*, *264*, 106711. https://doi.org/10.1016/j.fishres.2023.106711

Kaandorp, M. L. A., Lobelle, D., Kehl, C., Dijkstra, H. A., & Van Sebille, E. (2023). Global mass of buoyant marine plastics dominated by large long-lived debris. *Nature Geoscience*, *16*(8), 689–694. https://doi.org/10.1038/s41561-023-01216-0

Okubo, A. (1971). Oceanic diffusion diagrams. *Deep-Sea Research and Oceanographic Abstracts*, *18*(8), 789–802. https://doi.org/10.1016/0011-7471(71)90046-5

Poulain, P.-M., Gerin, R., Mauri, E., & Pennel, R. (2009). Wind Effects on Drogued and Undrogued Drifters in the Eastern Mediterranean. *Journal of Atmospheric and Oceanic Technology*, *26*(6), 1144–1156. https://doi.org/10.1175/2008JTECHO618.1

Révelard, A., Reyes, E., Mourre, B., Hernández-Carrasco, I., Rubio, A., Lorente, P., Fernández, C. D. L., Mader, J., Álvarez-Fanjul, E., & Tintoré, J. (2021). Sensitivity of Skill Score Metric to Validate Lagrangian Simulations in Coastal Areas: Recommendations for Search and Rescue Applications. *Frontiers in Marine Science*, *8*, 630388. https://doi.org/10.3389/fmars.2021.630388

Seo, S., & Park, Y. G. (2020). Destination of floating plastic debris released from ten major rivers around the Korean Peninsula. *Environment International*, *138*(March), 105655. https://doi.org/10.1016/j.envint.2020.105655

van der Mheen, M., van Sebille, E., & Pattiaratchi, C. (2020). Beaching patterns of plastic debris along the Indian Ocean rim. *Ocean Science Discussions*, 1–31. https://doi.org/10.5194/os-2020-50

van Sebille, E., Aliani, S., Law, K. L., Maximenko, N., Alsina, J., Bagaev, A., Bergmann, M., Chapron, B., Chubarenko, I., Cózar, A., Delandmeter, P., Egger, M., Fox-Kemper, B., Garaba, S. P., Goddijn-Murphy, L., Hardesty, D., Hoffman, M. J., Isobe, A., Jongedijk, C., … Wichmann, D. (2020). The physical oceanography of the transport of floating marine debris. *Environmental Research Letters*. https://doi.org/10.1088/1748-9326/ab6d7d

Vogt-Vincent, N. S., Burt, A. J., Kaplan, D. M., Mitarai, S., Turnbull, L. A., & Johnson, H. L. (2023). Sources of marine debris for Seychelles and other remote islands in the western Indian Ocean. *Marine Pollution Bulletin*, *187*, 114497. https://doi.org/10.1016/j.marpolbul.2022.114497

Zhang, Z., Wu, H., Peng, G., Xu, P., & Li, D. (2020). Coastal ocean dynamics reduce the export of microplastics to the open ocean. *Science of the Total Environment*, *713*, 136634. https://doi.org/10.1016/j.scitotenv.2020.136634

---

## Referee Comment (RC2)

[referee-annotated manuscript omitted]

---

## Referee Comment (RC3)

I thank the authors for their detailed response to my review. The purpose of the manuscript is now clearer to me, and I agree with the authors that a well-documented and transferable modelling system including beaching would be a useful contribution to the modelling community. This resolves one of my major concerns about the manuscript. I agree with most of the authors' responses to most other points raised in my review, and look forward to reading the revised manuscript.

However, I am writing this pre-emptive comment as a few points in my review may not have been made clearly enough, and I would like to clarify a couple of (significant) remaining concerns before the manuscript is revised.

Most importantly, I am still not convinced by the manuscript's claim that beaching scenario 2 is an improvement over beaching scenario 1. This is implied at several points in the manuscript, and is stated explicitly in lines 551-559. The only justification for this claim appears to be Figure 7, based on the beaching locations in scenario 2 conforming best with the real coastline (which is obvious, because scenario 2, by definition, only allows beaching at the real coastline – this outcome is predetermined). However, the purpose of this model is presumably not to produce something that *looks* realistic, but rather something that has skillful and useful predictive capacity (e.g. predicting accumulation hotspots). Predicting that beaching occurs at the real coastline is not useful, since we already know that. I do not think it is obvious that scenario 2 would have improved performance for predicting things stakeholders would be interested in (e.g. accumulation hotspots), because the 'real' coastline is not consistent with the hydrodynamic model grids.

For example, it is clear by comparing the top-right of Figs 7(b) and (d) that the CMEMS-IBI grid has ocean cells that intersect with the 'real' coastline. The hydrodynamics of CMEMS-IBI are blind to the 'real' coastline, however, so particles can travel into the 'real' coastline despite following nondivergent flow. Under scenario 2, where particles beach as soon as they reach the 'real' coastline (and setting aside the effects of Stokes drift), if there were, say, a NE-ward along-shore current, this would result in a convergence of particles beaching (as particles are being carried into the coast by the currents). This behaviour is not physically meaningful, as from the perspective of the hydrodynamic model (and therefore the underlying physics), the particles are not converging against the coast.

[Figure]

If I were going about evaluating these beaching scenarios, I would plot the density of beached particles per unit length of coastline, along the coast. It is obvious that scenario 2 will generate beaching locations that conform well with the coastline. It is not obvious that scenario 2 can predict which areas are high and low risk for beaching debris. It is of course entirely up to the authors how they wish to compare these beaching scenarios, but I do not see how the manuscript, in its current form, can make a justified recommendation about which scenario is 'best'.

The other clarification I wanted to make was on the diffusive parameterisation. I did not intend to question the use of $K = 10$ m$^2$ s$^{-1}$ for the IBI-CMEMS grid. My question is why the same value of $K$ was used for the finer (coastal and harbour) grids, which should have a much lower value of $K$, or none at all?

---

## Author Comment (AC2)

**Hernandez *et al.* Review**

**General comments**

Hernandez *et al.* introduce a modelling framework (LOCATE) for simulating marine dispersal in coastal regions using nested hydrodynamic grids. The authors apply LOCATE to a nested model of the Barcelona coastline (forced by a regional CMEMS model), and compare trajectory forecasting skill and beaching behaviour between the (higher resolution) nested model and (lower resolution) regional model, as well as the sensitivity of beaching to three beaching parameterisations.

Although LOCATE has been used in previous studies, this manuscript represents a considerable effort to document, describe, and illustrate a potential application for the framework. This is a great example of 'open science', and the quality of figures is also very high. Unfortunately, in its current form, the manuscript suffers from a lack of purpose and clarity, and feels more like a detailed supplement to Castro-Rosero *et al.* (2023). If the main purpose of the manuscript is to facilitate the wider adoption of LOCATE, the manuscript should clearly explain the advantages of LOCATE over just using Parcels directly, since the nested grid capability of Parcels is already quite accessible (i.e. if I were to investigate dispersal across nested grids, it is not clear to me why I would use LOCATE rather than 'pure' Parcels). If the main purpose is to instead explore the utility of nested grids for particle tracking in coastal waters, more thorough validation is needed than comparison with a single drifter profile. I realise that there is limited observational data available given the small domain size (using drogued drifters may help), but many of the manuscript's claims about "model accuracy" are weakly supported by evidence, and the manuscript therefore provides limited insights into the (dis)advantages of using nested grids beyond "there are differences". I also have concerns about some aspects of the methods, and other aspects were very difficult for me to understand.

If the authors can address the general comments above, then I would recommend major revisions to the manuscript (please see the attached document for specific and technical comments).

First of all, we wish to thank the referee for their extensive review and comments which have raised some very valid points.

The purpose of LOCATE is to provide a system to facilitate the study of the dispersion of marine debris in small-scale coastal settings. To achieve this, the use of nested hydrodynamic grids with high-resolution data is applied, and although this could be done using Parcels alone as suggested by the referee, the main purpose of LOCATE's methodology is to do so in conjunction with an appropriate method for detecting when and where particles arrive on land with as much precision as possible from the hydrodynamic data available and often given the complex coastal geometry. Given that coastal areas can experience very high amounts of beaching, special emphasis has been placed on the detection of the beaching of particles. This is especially relevant for areas such as the Barcelona coastline described in this study characterized by a complex geometry with a major harbour and several marinas (Port Olimpic, Forum), groynes, breakwaters and so on. Barcelona experiences discharge of debris on the coastline from various sources, such as rivers or other discharge outlets commonly found in urbanised coastal areas where debris is transported and interacts with the described complex geometry.

A beaching module based on a distance-to-shore parameter that detects the pre-calculated real shoreline by using distance data within grids during the simulation is included in LOCATE, which could be adapted to other areas where high-resolution coastline data may be available. The beaching module developed within LOCATE allows for a precise determination of which areas can be more

affected by particle beaching. The beaching module adapts to each computational grid. In small-scale studies, such as this one, such precision and highly granular information which can be determined in the post-simulation analysis when using the distance-based beaching module, is pertinent to being able to determine which areas, such as specific beaches or port structures, could be more at risk of receiving debris.

In the present work, it is shown that LOCATE's use of high-resolution hydrodynamic data that resolves coastal processes in conjunction with the distance-to-shore beaching module, allowed for substantially greater precision in terms of the measurements of where particles became beached. The reason for this is that by using these data together, complex geometric structures can be solved to a much greater extent than using the IBI-CMEMS grid alone where such structures are not taken into account or 'seen' due to the grid's coarse resolution. Furthermore, using the current velocity as a land detection parameter to determine when and where a particle becomes beached has been shown in this study to be insufficient and imprecise at localised scales, even if this has been widely used by other studies, albeit at much larger, even global scales.

The LOCATE model offers areas of functionality tailored towards use for coastal areas that would require substantial time investment and programming using Parcels from scratch. As mentioned above, the distance-to-shore beaching module can be used as a tool to determine where exactly particles become beached independently of hydrodynamic data resolution, while recreating real-life scenarios such as the continuous release of particles by a river or other discharge outlet can be easily achieved with relatively minor configurations. LOCATE can also be easily configured to track particle dispersion of one-off releases, such as runoff events, or even track drifters or tracers. Since there are high-resolution circulation data available throughout the Spanish coastline via the Spanish Port Authority (Puertos del Estado), LOCATE could be easily applied for similar studies in any of those areas with relative ease. Although Parcels is the engine behind the LOCATE model, it lacks the necessary considerations and requirements, which are included within LOCATE, to provide precise simulations of marine debris at localised coastal scales.

The concern about the model's accuracy is duly noted and in return, what we propose is that the model can be used for more precise measurements of particle beaching at localised settings. The abstract and conclusion have been edited to reflect this change. The model's validation has been extended with data on two more drifters that serve to increase the model's confidence to be used in coastal settings, where more consistent good skill scores were achieved when using the nested grids than using only the IBI-CMNEMS grid. However, the use of nested grids alone is not the sole focus of LOCATE as outlined above.

It is hoped that the concerns shown in the comments by the referee are addressed in the above statement and in the response to each of their points. Far from being a "detailed supplement" to other published works which have also used LOCATE to simulate the dispersion of particles, this work sets out to present a complete outline of the purpose and functionality available in LOCATE for use in coastal areas which may be affected by high amounts of beaching of debris.

With this in mind, the abstract (below), section 1.4 (objectives) and conclusion have been modified to reflect the points made above to offer greater clarity regarding the purpose of the study and the objectives. To avoid further confusion and add clarity the title has been shortened to:

LOCATE v1.0: Numerical Modelling of Floating Marine Debris Dispersion and Beaching in Coastal Regions Using Parcels v2.4.2

**Specific comments**

1. The abstract is long compared to the number of key messages from the paper. I would recommend condensing the abstract to make it easier for readers to identify the main points.
   - Reword abstract to:
     The transport mechanisms of floating marine debris in coastal zones remain poorly understood due to complex geometries and the influence of coastal processes, posing difficulties in incorporating them into Lagrangian numerical models. The numerical model LOCATE overcomes these challenges by coupling Eulerian hydrodynamic data at varying resolutions within nested grids using Parcels, a Lagrangian particle solver, to accurately simulate the motion of plastic particles where a high spatial coverage and resolution are required to resolve coastal processes. A particle beaching module was developed within LOCATE to address the detection of particles that cross the land-water boundary at coastal scales, using the pre-calculated distance of particles to the shoreline using high-resolution shoreline data during the simulation. This module displayed the highest precision of spatiotemporal beaching patterns relative to the real shoreline when compared to land detection mechanisms using solely hydrodynamic data in a beaching sensitivity analysis. Nested grids performed better than a coarse-resolution grid when analysing the model's dispersion skill by comparing drifter data and simulated trajectories. Another hydrodynamic grid configuration comparison applied the same observational debris discharge data as simulation input from two rivers around the Barcelona coastline, and the distance-based beaching module. High variability of beaching amounts between corresponding demarcated coastal areas was observed between simulations suggesting variations were influenced by coastal processes being resolved when using nested grids, with overall beaching levels >92% in each simulation. The model's ability to resolve complex coastal geometrical structures using nested grids was also demonstrated based on the particle residence times in areas of intricate shoreline configuration that were otherwise undetectable using the coarse-resolution grid. LOCATE can effectively integrate high-resolution hydrodynamic data within nested grids to model the dispersion of particles at coastal scales and represent deposition patterns with greater precision of particle beaching locations using high-resolution shoreline data.

2. Section 1.2 covers the dynamics of beaching debris in great detail, but most is of little relevance to the paper (since most of these dynamics are not investigated). I would recommend condensing.
   - Lines 55 -60 has been removed altogether and the remaining paragraph on beaching can then be integrated into the previous section

3. Section 1.4 begins by stating that the "goal of the present work is to simulate particles taking into account coastal processes using nested hydrodynamic grids at varying resolutions". I

would argue that this is the *method*, not the *goal*. As stated in the general comments, I would recommend clarifying what the actual goal of the paper is.

> o Change lines 88-89 to:
>
> The current study aims to present the functionality of a nested grid approach using high-resolution hydrodynamic data in conjunction with a particle beaching module that uses a distance-to-shore-based detection of the real shoreline, can resolve coastal processes and complex geometric structures at localised scales to better represent particle deposition patterns, accumulation and hot spots.

4. Section 2.2: What is the frequency of the Eulerian model output used in LOCATE? Line 159 states that "data are calculated daily" – daily strikes me as very low compared to the timescales over which currents resolved by a 2.5 km model (and certainly 350/70 m) change.

> o Modified the sentence on line 159 to:
>
> Coastal and harbour grids use the numerical model based on the Regional Ocean Modelling System (ROMS) (ROMS 2022, Shchepetkin, 2005). Coastal simulations with an hourly data frequency that use data from metocean operational products are nested into the IBI-CMEMS forecast solution using the SAMOA system (Alvarez-Fanjul, et al., 2018, Garcia-Leon et al., 2022, Sotillo, et al., 2015).

5. I cannot see any reference to tides in the manuscript outside of the introduction. Were tides not included in these simulations and, if not, why?

> o The Barcelona coastline is microtidal so tides are not considered, this has been added to the manuscript.

6. Section 2.3: Why was model output interpolated to a 'regular' grid (presumably A-grid)? As described in the OceanParcels documentation[1], A-grids introduce interpolation problems whereby particles get artificially stuck at the model coastline, because both velocity components approach zero at the coastline (i.e. particles will always stagnate at the coastline of an A-grid, regardless of the strength of any along-shore currents). Particularly given the focus of this manuscript on beaching, I am concerned about the potential for beaching to occur due to interpolation artefacts, rather than any physical reason. The authors could consider using the original C-grids instead, or the free-slip boundary condition that was recently implemented in Parcels[2].

> o This was a mistake on my part, and I apologise for the confusion. Line 181 can be changed to:
>
> "Circulation data as numerical simulations available from PdE through the OPeNDAP server were provided in A-grids. Although the IBI-CMEMS grid also had an A-grid configuration, further configuration was necessary due to the coastal and harbour grids being oriented towards the coastline, having areas which contained no data. To overcome this, an interpolation between the three grids that filled out empty values in the grids with higher resolution with the equivalent spatiotemporal data from the lower resolution grids in these points was carried out using the
* * *
[1] https://docs.oceanparcels.org/en/latest/examples/documentation_stuck_particles.html

[2] https://docs.oceanparcels.org/en/latest/examples/documentation_unstuck_Agrid.html#3.-Slip-boundaryconditions

UPC_resample_datasets script. The result of this interpolation can be seen in Fig.2a."

The interpolation from a C-grid to an A-grid configuration happens before the data is served by the OPeNDAP server. Therefore, the data available from the Spanish port authority (PdE) through the OPeNDAP server is not available in C-grid configuration. However, using nested regular grids does make applying them in Parcels a lot easier. At UPC we also perform Hydrodynamic numerical simulations and we indeed collaborate with Puetos del Estado to optimize the OPeNDAP data information therefore we are able to perform simulations with C-grid. However, such simulations are performed in large computing machines and are not accessible to everyone. Therefore, we have decided to keep the model accessible through the use of OPeNDAP.

Regarding difficulties with A-grids such as particles becoming stuck, the particles are deleted from the simulation when it is determined that they reach land, which is one of the solutions outlined in the Parcels document provided to prevent this. In fact, the beaching parameters take full advantage of this functionality, as well as when particles are exported so they do not become 'stuck' on the grid boundary. The deleting of particles is described in line 260. The following line has been added to the end of section 2.3:
"Artificial stagnation of particles on the coastline, which can be a concern when using A-grids was circumvented by deleting the particles on crossing a predetermined land-water boundary as described in section 2.6."

7. Section 2.5: Given that LOCATE uses a constant value for $K$, most of the detail in lines 233-243 is unnecessary and should be removed. I am also confused by the choice of a constant value for $K$ (10 m$^2$ s$^{-1}$). Should this not be scale-dependent? Okubo (1971) would suggest that a value of 10 m$^2$ s$^{-1}$ would be appropriate to reproduce the effects of unresolved scales of motion at a resolution of around 10 km, much coarser than the resolution used in this study.

   o Similar studies using similar resolution scales use a Kh value of 10m$^2$s$^{-1}$ mentioned in Okubo 1971:
   Onink, V., Jongedijk, C. E., Hoffman, M. J., van Sebille, E., & Laufkötter, C. (2021). Global simulations of marine plastic transport show plastic trapping in coastal zones. *Environmental Research Letters*, *16*(6). https://doi.org/10.1088/1748-9326/abecbd

   Onink, V., Kaandorp, M. L. A., Van Sebille, E., & Laufkötter, C. (2022). Influence of Particle Size and Fragmentation on Large-Scale Microplastic Transport in the Mediterranean Sea. *Environmental Science and Technology*, *56*(22), 15528–15540. https://doi.org/10.1021/acs.est.2c03363

   Citing the Okubo 1971 paper and using 10m$^2$s$^{-1}$
   Liubartseva, S., Coppini, G., Lecci, R., & Clementi, E. (2018). Tracking plastics in the Mediterranean: 2D Lagrangian model. Marine Pollution Bulletin, 129(1), 151–162. https://doi.org/10.1016/j.marpolbul.2018.02.019

Under the section of "Dispersal model"

Lacerda, A. L., Rodrigues, L. D., Van Sebille, E., Rodrigues, F. L., Ribeiro, L., Secchi, E. R., Kessler, F., & Proietti, M. C. (2019). Plastics in sea surface waters around the Antarctic Peninsula. *Scientific Reports*, *9*(1), 1-12. https://doi.org/10.1038/s41598-019-40311-4

Since this is used in studies that use either the same or similar resolution to the IBI-CMEMS grid we see no reason to use a different value, especially since there are no data on diffusivity values in this region. Values $5m^2s^{-1}$ to $10m^2s^{-1}$ were not found to materially affect where particles end up in preliminary simulations. The paper for Okubo 1971 will be cited accordingly.

8. Section 2.7: I am surprised by the decision to use undrogued drifters. Drifters lacking a drogue experience a significant direct push from the wind due to the exposed part of the drifter (Poulain et al., 2009), which is not included in LOCATE. Although the authors' concern about velocity shear within the upper water column is valid, there is some evidence that surface currents reasonably represent the forces driving drogued drifter movement (Imzilen et al., 2023). Using drogued drifters may also increase the number of drifters available to the authors for use in validation.

   o In Parcels virtual particles by default do not have size, shape or buoyancy defined, which for these simulations is not an issue since the particles are assumed to be floating just beneath the surface. This is mentioned in line 518 as to why wind drag data is not included. Stokes drift data is considered with the IBI-WAV data, which has been shown to have a considerable effect on drifter trajectories, especially over large distances. Moreover, laboratory experiments have shown that floating particles move with Stokes drift velocity unaffected by particle size and density (Alsina, et al 2020).

   o Even though in the article it is mentioned that only drogueless drifters are used, this term was imprecise and will be changed to "drifters with drogues < 1 m" which includes CODE drifters used in this study. The decision to use this limitation on the drifter drogue depth was made to take Stokes drift into consideration.
   Thus, line 266 has been changed to:
   "Drifter data were selected on the condition of having drogues < 1 m to assess only the influence of surface currents, including Stokes drift, as it is assumed more realistic for a floating particle."

   o In the provided article by Imzilen et al 2023 they hypothesided that the impact of Stokes drift on dFAD trajectories would be minor given their considerable subsurface structure and transport patterns that closely resemble those of drogued drifters. In our case we are considering virtual particles just beneath the surface as a proxy for floating macrolitter, thus using data from drifters with drogues of typically 15 m would not have provided with an appropriate comparison given that Stokes drift is reduced at that depth, and we do include Stokes drift data on the simulations.

9. Section 2.7: I am uncomfortable with the use of the word "validation" for the comparison with a single drifter. Is this really enough data to count as validation?

   o There were data for 2 other drifters that were not included in the article because even though their trajectories did cross the area where the high-resolution grids apply, they crossed the area where the coastal grid applied (resolution of 350 m ) but

not where the harbour grid applied (resolution 70 m). The drifter that was featured, however, did benefit from crossing all grids. We hope that having the trajectories of 3 drifters is sufficient for a preliminary work while addressing the need for more data (as mentioned in the discussion). The data for the other 2 drifters and their skill scores have been included in the following figure:

[Figure]

o Line 269 has been changed to:

"Validation simulations were conducted for the available drifters. CODE drifter 6592, deployed in February 2022 was chosen as the most suitable because its trajectory crossed the coastal and harbour grids for a period long enough to analyse the skill of the model to forecast the trajectory 6, 24, or 72 h ahead, compared as a function of when the forecast started. CODE drifters 6607 and 6608 only crossed the coastal domain and were released within a minute of each other in March 2022."

o Line paragraph from 274 in the method has been changed to:

"For drifter 6592, simulations were conducted from the point where the trajectory transected the coastal grid boundary, at coordinates 41.18162°N and 2.24084°E. The provided data did not include timestamps, only a deployed and end date, so the drifter trajectory was temporally interpolated to provide hourly data points where 100 particles were released at every step. Particles were released between the period 9 March 2022 18:11:00 to 14 March 2022 18:11:00. The number of particles was determined through a sensitivity analysis of the standard deviation using varying amounts of particles (Castro-Rosero et al.,2023). Simulations for drifter 6607 were conducted at coordinates 41.19657°N and 2.27386°E from the period 11 March 2022 09:14:00 to 14 March 2022 22:14:00, and for drifter 6608 from coordinates 41.20218°N and 2.28279°E from the period 12 March 2022 07:13:00 to 15 March 2022 09:13:00. The same simulations were conducted using nested grids and using the IBI-CMEMS grid only."

o Added this paragraph after line 383 in Results:

"As seen in Fig.(plot above), the SS of drifter 6652 was much better when using the nested grids than only the IBI-CMEMS grid (SS=0.53 compared to SS=0.07). For drifter 6608 the IBI-CMEMS grid performed slightly better (SS=0.74 compared with SS=0.68). Qualitatively, it can be observed that the particle trajectories using the IBI-CMEMS grid were being displaced towards the coastline with substantial beaching of particles, while the particle trajectories with the nested grids moved further out to sea, with the real drifter trajectories somewhat in the middle. In both cases, the difference in SS is only 0.06 which can be taken as being minimal."

o Lines 441 to 449 in the discussion has been changed to:

"This is indicative of the challenges associated with predicting trajectories close to the shoreline influenced by coastal processes, amplified by the strong influence of the alongshore northern current as seen south of the Barcelona city area in Fig.6g to Fig.6j and in the trajectories in Fig 7 (above plot). There is a notable difference in how the different grids performed during the dates of the simulation, with the IBI-CMEMS grid showing the northern current much closer to the coastline than the higher-resolution grids, with a greater probability for beaching as seen for drifter 6607 (Fig.7(h)). The consistently good and generally better SS values of the simulated drifter trajectories when using the nested grids compared to more variable results when using the IBI-CMEMS grid, demonstrate that the model can be effectively validated with the nested grids producing generally more favourable results. Drawing direct comparisons between nested grids and the IBI-CMEMS grid using the SS test, however, is challenging. Within a single trajectory, the influence and contribution of each grid as a particle moves across different domains cannot be isolated due to the

cumulative nature of the test, even if it is possible to numerically determine which grid has provided the hydrodynamic data for that time-step. Additionally, an area of future work would be to address the paucity of available and suitable drifter information at coastal scales where high-resolution hydrodynamic data may apply, to increase the statistical robustness of additional validation analyses."

10. Section 2.7: How were drifter trajectories temporally interpolated if there are no timestamps associated with locations?

   o A linear interpolation was the most prudent way to do this. Changed line 275 to:

   "The provided data did not include timestamps, only a deployed and end date. It was assumed, however, that the drifter was fully operational and recording data at regular intervals, thus a reconstructed trajectory using the coordinates and the number of data points available were linearly interpolated to provide hourly data points, from which 100 particles were released at every step."

11. Section 2.8: I cannot find a precise definition for the residence time in this manuscript – please clarify how this was calculated.

   o Added the definition to line 27:

   "(the time it spends in a region of interest)"

12. Section 2.8: Since the manuscript does not investigate or validate the temporal variability of debris accumulation, I do not see the point of varying the input rate of debris.

   o Since observational data were available from the Schirinzi et al 2020 article, the decision was made to use a real-life scenario to test LOCATE's capabilities for simulating the beaching of particles on a real shoreline based on the distance to shore beaching parameter, and test high-resolution data to solve complex geometric structures in the study domain. The analysis of the main simulation in this article using nested grids comparing to hydrodynamic conditions and beach cleanup data is being prepared for another publication. Including that side of the analysis would have been too much and taken away from the focus of this work.

13. Section 2.9: If I understand correctly that scenario 2 defines the distance-to-shore relative to the true (rather than model) coastline, would this not make it impossible for debris to beach in many places? There are presumably parts of the model grid where ocean cells do not quite reach the true coastline (see below), so it is never possible for the distance-to-shore parameter to reach zero here, and therefore also impossible for particles to beach? In any case, I do not understand the point of using the real coastline in a beaching parameterisation – particles within the simulation do not 'know' anything about the real

coastline, they only 'know' about the model coastline.

[Figure]

True land
Model coast
Unbeachable coast

- o While it is true that there are some cells with no data as seen in Fig.2, the interpolation performed by Parcels with land data (zero values) means that these cells will have interpolated velocity data. This can be seen in the beaching patterns in Fig 7b when all the beaching occurs on land cells not at sea cells.
- o The point of using the real shoreline with a distance-to-shore beaching parameter was to avoid such beaching patterns that could make identifying exactly when a particle crosses the land-water boundary impossible at small scales such as this study. Given that all cells around the shore have velocity data given by the grids or by interpolation with zero-value adjacent land cells, having the beaching parameter set to "when the distance to the shoreline is less than zero" gives a much better indication of where exactly a particle crosses the boundary than simply using the current velocity as a land detector, especially where the resolution of the IBI-CMEMS grid applies, which is 2.5 km per cell. Therefore, particles do 'know' where the real coastline is because this distance-to-shore data is used in the simulation as nested distance grids in a Fieldset. together with the hydrodynamic grids in another Fieldset while conducting the simulation. This is explained in the paragraph in line 336.
- o Clarified in line 336 to:
  "Given that kernels in Parcels are limited to basic arithmetic operations and conditions, Parcel's interpolation capabilities were utilised to calculate the real-time minimum distance between particle and shoreline for scenarios 2 and 3 with distance data available in a fieldset, so that particles can effectively detect the real shoreline given a beaching parameter."
- o Changed line 536 to:
  "The beaching pattern in Figure 7b shows that there are no areas around the coastline that are left uncovered by data, and that beaching always occurs on land. Even if some cells around the coastline do not have velocity data as seen in Fig. 2, Parcels' interpolation capabilities use the zero velocity values from adjacent land cells to interpolate velocities of shoreline cells without velocity data."

14. Section 2.9: I do not understand scenario 3. Lines 533-535 makes it sound like particles beached if they travelled less than 1.694 km in 6 hours, but I am not sure if this is correct.
- o That is correct. While the scenario may not seem realistic, it is a scenario created to reflect the time dependency of some beaching parameters used in other studies,

whether deterministic or probabilistic. This scenario assumes that all particles become beached eventually if they get close enough to land for a minimum amount of time.

15. Section 3.1: I did not understand this section (particularly what was meant by 'horizons') until I read Révelard et al., (2021). Please clarify (probably in the methods) the methods, i.e. comparing the skill of the model to *forecast* the trajectory of a drifter 6/24/72 h ahead, as a function of when the forecast started.

    o Mentioned in point 9 as an amendment to line 269

16. Figure 6: How is the 'mean trajectory' defined? Is this just the arithmetic mean of all latitudes and longitudes at a point in time?

    o That is correct.

17. Figure 7: Please use a divergent colourmap for these figures, centred at 0. The use of a sequential colourmap makes it very difficult to tell which particles are close to the coastline, versus which have over/undershot.

    o Understood and agree, the figure has been modified to this effect.

18. Section 3.2: Please specify which beaching scenario was used for Figure 9 and lines 403-418.

    o Scenario 2 was used for all simulations after the beaching sensitivity analysis and was used for the results in Fig.8 and Fig.9. The use of beaching scenario 2 will be mentioned in lines 395 and 403.

19. Figure 8: The left and right panels do not seem to correspond in this figure. For example, lots of blue particles appear in the far SW corner on May 31 (left), but not in the panel on the right.

    o The images do correspond even if the patterns may differ slightly. This is because the plots on the right are heatmaps of particle concentrations (num of particles/km$^2$) and while there may be particles showing blue dots on the left side plots, the concentrations may be low and not be enough to show on the heatmaps. That is why both plots were shown for comparison.

20. Section 4.1: Is it not surprising that the regional grid did not perform considerably worse than the nested grid? This is one of the most interesting observations in the manuscript for me (that the regional grid often had similar performance to the nested grid), and seems to somewhat contradict the manuscript's conclusion that using nested grid improves the accuracy of predictions (although there is insufficient validation in the manuscript to be sure).

o Agreed, it is not surprising that the regional grid does not perform considerably worse than using the nested grids, as outlined in line 158, "Coastal and harbour grids use the numerical model based on the Regional Ocean Modelling System (ROMS) and data are calculated daily using coastal simulations using data from metocean operational products nested into the IBI-CMEMS forecast solution using the SAMOA system… "

o However, it seems that the focus is slightly misunderstood and for that, the abstract has been edited. This study looks at the performance of IBI-CMEMS/nested grids with regard to beaching as well as forecasting trajectories. The validation using 3 drifter buoys showed somewhat better results using the nested grids. What we argue is that using IBI-CMEMS alone at small scales does not solve complex geometric

structures, and thus cannot provide accurate beaching accumulation hot spots. Complex structures are missed completely, as seen in Fig.9 where the port area registers particle residence times 18 times less than when using high-resolution data. The accuracy of the predictions is in the context of where beaching occurs at small scales aided by the high-resolution data that considers the complex geometry of the coastline. This is mentioned in line 480:

"Solely relying on large-scale, low-resolution grids such as IBI-CMEMS is insufficient for coastal-scale simulations, although some limitations exist in the high-resolution hydrodynamic data utilised in this study"

21. Section 4.2: Line 540 states that, under scenario 1, particles can travel "several kilometres inland before being considered beached". Along similar lines to my comment on Section 2.9, does this really matter? Is it really a problem that beaching locations are 'wrong' by a few kilometres in a model with a resolution of a few kilometres?

   o It does matter very much at local and coastal scales, which is precisely the problem that this study aims to highlight. This study is localised around the Barcelona coastline and therefore having a model that predicts that debris becomes beached several kilometres inland renders it useless in knowing which beaches in or around the city, would be more affected by debris being released by the surrounding rivers after a rainfall event. This could have management implications for how/when cleanups are conducted. This is a small-scale study which differs from the global scale or Mediterranean scale studies of a similar nature done so far. Hence, why the distance-to-shore beaching module was developed.

22. Section 5: The manuscript does not validate any beaching predictions, so there is no data to support the claim that "using real-time particle distance to the shoreline… can accurately model particle arriving time and beaching locations". Similarly, no data has been presented to support the claim that "LOCATE… provides accurate depictions of accumulation zones and debris hot-spots…", at least not in this manuscript.

   o The results show how complex structures are completely missed when only using the IBI-CMEMS grid, as described in point 21. Therefore, using the distance-to-shore beaching module and high-resolution hydrodynamic data can give greater confidence in the accuracy of the particle beaching locations when these data are applied. This point is made in the edited conclusion.

23. Table A1: This table may not set out to be exhaustive but, in case of interest (as this is quite a nice compilation of beaching parameterisations that could be useful for others), some other studies using 'deterministic' parameterisations include Bosi et al., 2021; Cardoso & Caldeira, 2021; Critchell & Lambrechts, 2016; Dobler et al., 2019; Seo & Park, 2020; Zhang et al., 2020. Some other studies using 'probabilistic' parameterisations include Kaandorp et al., 2023; van der Mheen et al., 2020; Vogt-Vincent et al., 2023.

   o Thank you - these have been added to table A1.

24. Did the authors observe any artefacts in particle concentrations along the nested domain boundaries? There are quite striking discontinuities in surface velocity along domain boundaries in Figure 2, and it would be interesting to know whether this causes any artificial particle convergence/divergence at these boundaries.

o Yes, there were instances where particles would converge at some places around the coastal and IBI-CMEMS boundary because of cells where the current velocity was especially slow at that moment in time. The particles would be kept drifting until velocity data in the cell at another moment in time would allow it to move along since they did not cross the land/water boundary. Any artificial convergence was temporary and sporadic. This was a limitation of the hydrodynamic data provided, thus out of our control. However, any effect this could have had on the residence times could be taken as minimal and would not have affected the areas of interest which was where the high-resolution data applied.

**Technical comments/corrections**

1. Lines 29-31: The number of references for the physics of marine dispersal is excessive in my view. I would recommend condensing, e.g. just using van Sebille et al., (2020).
   - Understood, changed as advised above.

2. Lines 31-33: It is not clear to me why this sentence is relevant. Lines 31-36 could be removed for brevity.
   - Understood. Both sentences were removed.

3. Line 40: This is a very specific range (50-600 m). Where is this range from?
   - This range is observed in the Catalan coastline, with the outer value of the range applying during stormy conditions. However, the range itself can be removed.

4. Line 42: Are density gradients really a *driver* of coastal currents (is the driver not the process that generated the density gradients in the first place, e.g. upwelling/downwelling)?
   - True, modified.

5. Line 45: The referenced paper by Stokes was published in 1880, not 2009!
   - Agreed, this has been changed. I quoted the print publication by Cambridge University Press which is the available version with the DOI, not the original date.

6. Line 70: Would recommend changing "high spatial discretisation" to "fine spatial discretisation".
   - Thank you. Modified.

7. Line 82: I do not follow how this relationship is exponential. Computational costs should scale broadly linearly with the area modelled, and a polynomial (not exponential) relationship with the model resolution.
   - Removed the word exponentially altogether.

8. Lines 160-161: Please separate the references for ROMS, IBI-CMEMS, and SAMOA.
   - Done.

9. Line 172: Typo ('downlaod')
   - Thank you, done.

10. Line 174-178: I would recommend giving the resolution of the wave model here (I assume 1/20 degree, based on line 494).
    - Thank you, done. It is 1/20º

11. Line 198: It is not clear what is meant by "typical stochasticity". I assume the authors meant subgrid scale diffusion.
    - Done.

12. Line 242-243: Citing OPeNDAP is unnecessary here (the text is describing the data itself, not OPeNDAP).

   ○ Done

13. Line 246: Please state or cite where the value (1/3) for the variance of the random process comes from.

   ○ The figure of ⅓ for the variance of random uniform probability distribution can be found in a number of studies:
   Table 1 on:
   Scutt Phillips, J., Sen Gupta, A., Senina, I., van Sebille, E., Lange, M., Lehodey, P., Hampton, J., & Nicol, S. (2018). An individual-based model of skipjack tuna (Katsuwonus pelamis) movement in the tropical Pacific ocean. *Progress in Oceanography*, *164*(February), 63–74. https://doi.org/10.1016/j.pocean.2018.04.007

   Equation 1 on:
   Ross, O. N., & Sharples, J. (2004). Recipe for 1-D Lagrangian particle tracking models in space-varying diffusivity. *Limnology and Oceanography: Methods*, *2*(9), 289–302. https://doi.org/10.4319/lom.2004.2.289

   Also in a more recent paper in Section 2.2, equation 1:
   Onink, V., Kaandorp, M. L. A., Van Sebille, E., & Laufkötter, C. (2022). Influence of Particle Size and Fragmentation on Large-Scale Microplastic Transport in the Mediterranean Sea. *Environmental Science and Technology*, *56*(22), 15528–15540. https://doi.org/10.1021/acs.est.2c03363

   Equation 1 in:
   Onink, V., Jongedijk, C. E., Hoffman, M. J., van Sebille, E., & Laufkötter, C. (2021). Global simulations of marine plastic transport show plastic trapping in coastal zones. *Environmental Research Letters*, *16*(6). https://doi.org/10.1088/1748-9326/abecbd

   Also, equation 2 in:
   Castro-Rosero, L. M., Hernandez, I., Alsina, J. M., & Espino, M. (2023). Transport and accumulation of floating marine litter in the Black Sea: insights from numerical modeling. *Frontiers in Marine Science*, *10*(August), 1–19. https://doi.org/10.3389/fmars.2023.1213333

14. Table 3: It is a bit confusing using the same numbering for sensitivity test and beaching scenario. It may be clearer to remove the sensitivity test numbering, and change sensitivity test 1 to test 1R (beaching scenario 1, river release), test 2 to 2R, test 4 to 1H, etc.

   ○ Understood, done.

15. Figure 6 caption, last line: I assume "(f) to (j) should be "(h) to (l)".

   ○ Yes, apologies for the oversight.

16. Lines 365-402: This paragraph is unrelated to beaching, so should not be in section 3.2.

   ○ Agreed. This was an oversight with the positioning of the figures etc. A subsection heading was created for lines 395 onwards
   3.3 Simulations of river release particles

17. Line 404-405: This line implies that no particles were retained (at sea, unbeached). Is this correct?

   o That is correct. By the end of the simulation, all particles had either become beached or were exported from the domain.

18. Line 406-408: Although it is possible to work this out from context, please specify that "The Prat de Llobregat area *had* 12.7% more particles…" means that 12.7% more particles *beached*. This sentence also makes it sound like these are relative percentages, whereas based on Figure 9 it looks like these are absolute percentages.

   o Rephrased line 408 to:

   "The Prat de Llobregat area received 12.7% more beached particles with the IBI-CMEMS simulation (24.0% with the IBI-CMEMS grid compared to 11.3% with nested grids), whereas the Llobregat River mouth showed 8.7% more beached particles with the nested grid simulation (43.5% with nested grids compared to 34.8% with the IBI-CMEMS grid)."

19. Line 431: Typo ("harbourss")

   o Thank you. Done

20. Lines 439-441: Please state for which configuration the SS value was higher (presumably the nested grid).

   o Correct, it was the nested grids. Clarified.

21. Line 483: I would recommend replacing "wave-induced currents" with "wave-induced Eulerian (mean) currents". The Stokes drift is not a current, but some readers might be confused.

   o Thanks, done.

22. Line 520: Please specify that "both simulations" refers to the regional vs nested grids.

   o That is correct, done.

23. Lines 570-571: The manuscript does not discuss the difference in computational cost between the regional and nested configurations, so I would recommend removing this sentence (or quantifying the difference in computational cost, and moving this to the discussion).

   o Agree, removed. What was meant was that using the distance-based grid for the distance-to-shore calculation did not noticeably affect the simulation time.

24. The labels "IBI-CMEMS" and "regional grid" are used interchangeably in this manuscript, which can get confusing. I would stick to one of them.

   o Agree, the nomenclature has been standardised.

**References**

Bosi, S., Broström, G., & Roquet, F. (2021). The Role of Stokes Drift in the Dispersal of North Atlantic Surface Marine Debris. *Frontiers in Marine Science*, *8*(August), 1–15. https://doi.org/10.3389/fmars.2021.697430

Cardoso, C., & Caldeira, R. M. A. (2021). Modeling the Exposure of the Macaronesia Islands (NE Atlantic) to Marine Plastic Pollution. *Frontiers in Marine Science*, *8*(April).

https://doi.org/10.3389/fmars.2021.653502

Critchell, K., & Lambrechts, J. (2016). Modelling accumulation of marine plastics in the coastal zone; what are the dominant physical processes? *Estuarine, Coastal and Shelf Science*, *171*, 111–122. https://doi.org/10.1016/j.ecss.2016.01.036

Dobler, D., Huck, T., Maes, C., Grima, N., Blanke, B., Martinez, E., & Ardhuin, F. (2019). Large impact of Stokes drift on the fate of surface floating debris in the South Indian Basin. *Marine Pollution Bulletin*, *148*(May), 202–209. https://doi.org/10.1016/j.marpolbul.2019.07.057

Imzilen, T., Kaplan, D. M., Barrier, N., & Lett, C. (2023). Simulations of drifting fish aggregating device (dFAD) trajectories in the Atlantic and Indian Oceans. *Fisheries Research*, *264*, 106711. https://doi.org/10.1016/j.fishres.2023.106711

Kaandorp, M. L. A., Lobelle, D., Kehl, C., Dijkstra, H. A., & Van Sebille, E. (2023). Global mass of buoyant marine plastics dominated by large long-lived debris. *Nature Geoscience*, *16*(8), 689–694. https://doi.org/10.1038/s41561-023-01216-0

Okubo, A. (1971). Oceanic diffusion diagrams. *Deep-Sea Research and Oceanographic Abstracts*, *18*(8), 789–802. https://doi.org/10.1016/0011-7471(71)90046-5

Poulain, P.-M., Gerin, R., Mauri, E., & Pennel, R. (2009). Wind Effects on Drogued and Undrogued Drifters in the Eastern Mediterranean. *Journal of Atmospheric and Oceanic Technology*, *26*(6), 1144–1156. https://doi.org/10.1175/2008JTECHO618.1

Révelard, A., Reyes, E., Mourre, B., Hernández-Carrasco, I., Rubio, A., Lorente, P., Fernández, C. D. L., Mader, J., Álvarez-Fanjul, E., & Tintoré, J. (2021). Sensitivity of Skill Score Metric to Validate Lagrangian Simulations in Coastal Areas: Recommendations for Search and Rescue Applications. *Frontiers in Marine Science*, *8*, 630388. https://doi.org/10.3389/fmars.2021.630388

Seo, S., & Park, Y. G. (2020). Destination of floating plastic debris released from ten major rivers around the Korean Peninsula. *Environment International*, *138*(March), 105655. https://doi.org/10.1016/j.envint.2020.105655

van der Mheen, M., van Sebille, E., & Pattiaratchi, C. (2020). Beaching patterns of plastic debris along the Indian Ocean rim. *Ocean Science Discussions*, 1–31. https://doi.org/10.5194/os-2020-50 van Sebille, E., Aliani, S., Law, K. L., Maximenko, N., Alsina, J., Bagaev, A., Bergmann, M., Chapron, B., Chubarenko, I., Cózar, A., Delandmeter, P., Egger, M., Fox-Kemper, B., Garaba, S. P., GoddijnMurphy, L., Hardesty, D., Hoffman, M. J., Isobe, A., Jongedijk, C., … Wichmann, D. (2020). The physical oceanography of the transport of floating marine debris. *Environmental Research Letters*. https://doi.org/10.1088/1748-9326/ab6d7d

Vogt-Vincent, N. S., Burt, A. J., Kaplan, D. M., Mitarai, S., Turnbull, L. A., & Johnson, H. L. (2023). Sources of marine debris for Seychelles and other remote islands in the western Indian Ocean. *Marine Pollution Bulletin*, *187*, 114497. https://doi.org/10.1016/j.marpolbul.2022.114497

Zhang, Z., Wu, H., Peng, G., Xu, P., & Li, D. (2020). Coastal ocean dynamics reduce the export of microplastics to the open ocean. *Science of the Total Environment*, *713*, 136634. https://doi.org/10.1016/j.scitotenv.2020.136634

---

## Author Comment (AC3)

I thank the authors for their detailed response to my review. The purpose of the manuscript is now clearer to me, and I agree with the authors that a well-documented and transferable modelling system including beaching would be a useful contribution to the modelling community. This resolves one of my major concerns about the manuscript. I agree with most of the authors' responses to most other points raised in my review, and look forward to reading the revised manuscript.

However, I am writing this pre-emptive comment as a few points in my review may not have been made clearly enough, and I would like to clarify a couple of (significant) remaining concerns before the manuscript is revised.

Most importantly, I am still not convinced by the manuscript's claim that beaching scenario 2 is an improvement over beaching scenario 1. This is implied at several points in the manuscript, and is stated explicitly in lines 551-559. The only justification for this claim appears to be Figure 7, based on the beaching locations in scenario 2 conforming best with the real coastline (which is obvious, because scenario 2, by definition, only allows beaching at the real coastline – this outcome is predetermined). However, the purpose of this model is presumably not to produce something that *looks* realistic, but rather something that has skillful and useful predictive capacity (e.g. predicting accumulation hotspots). Predicting that beaching occurs at the real coastline is not useful, since we already know that.

- The main is purpose is not to predict that beaching occurs at the coastline, but to predict where beaching occurs along the coastline with greater certainty and precision. For example, using the real coastline we can determine which beaches around the Barcelona coastline experience greater amounts of particle beaching after discharges from a heavy rainfall event. If we used the coastline as per the hydrodynamic grids, even with a maximum resolution of 70 m, it would be difficult to determine where exactly beaching is more likely to occur, let alone with coarser resolutions. Using the distance-to-shore data, small-scale geometric structures are considered especially if they are in an area where high-resolution hydrodynamic data is being used. Using these data can produce results which are of more practical use at small and localised scales, such as the present study.

- Below is a closeup plot of how the hydrodynamic grids sees the Barcelona city area with Scenario 1 (detection using hydrodynamics). The port grid data (resolution of 70 m) applies to most of this image except the bottom left corner below 41.3N where the coastal grid data applies (resolution of 350 m). Even though the coastal grid is still high resolution, the beaching pattern is quite jagged. Where the port grid applies a lot of the structures such as piers and groynes and even areas within the port itself are missed out. Please see the image below in a closeup of the beaches to the north see this point in more detail

[Figure]

-

- This is a closeup from the image above of the area above 41.36N, with a closeup view of the Barcelona beaches using the hydrodynamic grids as detection where the highest resolution hydrodynamic data applies. Structures, individual zones or beaches are difficult to distinguish or not at all.

[Figure]

- Below is the same area using the distance-to-shore parameter. The coastline, including port structures, beaching patterns are much more closely matched in terms of the real coastline.

[Figure]

-

- Closeup of the Barcelona beaches using the distance-to-grid parameter. Beaches, and small-scale structures are "seen" and more precise measurements in these areas can be conducted. Additionally, further parameterisations more specific to beach-scale dynamics could be developed with this system as areas of future research.

[Figure]

- With the above comparison, it is hoped that in discerning potential accumulation zones at a local scale, having a distance-to-shore parameter to determine beaching shows more reasonable patterns at the very least and may be of more practical use in beach debris management.
- We focus on small or localised scales instead of larger scales where such detail is not required and having such precision is no longer meaningful.
- We are doing further work using a further nested grid within the 70 m port grid to explore coastal processes in more detail. This grid would have a resolution of 14 m.
- Using the beaching module with the hydrodynamic grids it would be very difficult, if not impossible, to quantify how much debris reaches specific beaches, for example.
- The caveats are discussed in the revised manuscript.

I do not think it is obvious that scenario 2 would have improved performance for predicting things stakeholders would be interested in (e.g. accumulation hotspots), because the 'real' coastline is not consistent with the hydrodynamic model grids.

- The sensitivity analysis highlights that the real coastline differs substantially from how the hydrodynamic grids "see" the coastline and this can have substantial effects on where particles are considered beached at small scales. The assumption that stakeholders may not be interested in how the real coastline is resolved in terms of particle beaching accumulation may be true for large-scale studies, where the real coastline may not even be relevant. However, we would argue that at small scales, the concerns and requirements can be quite different and knowing which areas are affected gains relevance, as shown in the closeup plots above.

For example, it is clear by comparing the top-right of Figs 7(b) and (d) that the CMEMS-IBI grid has ocean cells that intersect with the 'real' coastline. The hydrodynamics of CMEMS-IBI are blind to the 'real' coastline, however, so particles can travel into the 'real' coastline despite following nondivergent flow. Under scenario 2, where particles beach as soon as they reach the 'real' coastline (and setting aside the effects of Stokes drift), if there were, say, a NE-ward along-shore current, this would result in a convergence of particles beaching (as particles are being carried into the coast by the currents). This behaviour is not physically meaningful, as from the perspective of the hydrodynamic model (and therefore the underlying physics), the particles are not converging against the coast.

[Figure]

- The example above shows alongshore current close to the coastline in an area where the low-resolution CMEMS grid applies. The resolution of 2.5 km in these areas mean that such coastal processes are not resolved hydrodynamically anyway. Therefore it would be very unlikely for particles to reach the real coastline that would otherwise be caught in a current resulting from a coastal process where low-resolution hydrodynamic data applies.

If I were going about evaluating these beaching scenarios, I would plot the density of beached particles per unit length of coastline, along the coast. It is obvious that scenario 2 will generate beaching locations that conform well with the coastline. It is not obvious that scenario 2 can predict which areas are high and low risk for beaching debris.

- Beaching along the coastline density plot for the homogeneous release in the beaching sensitivity analysis hydrodynamic grid as detection (above) and distance-to-shore as detection (below). Artefacts are shown in red, the area where high-resolution hydrodynamic data is shown in the dashed orange rectangle. One pixel represents approx 1km$^2$.

[Figure]

[Figure]

-
- Aside from the qualitative differences between using the hydrodynamic grids (above) and distance-to-shore grid (below), there are artefacts (red circle) when using the hydrodynamic grids also visible in Fig7a and b in the manuscript. At 41.6N and 2.6E there is a hotspot at the intersection of the edges of the hydrodynamic boundary, which is much more prominent when using the hydrodynamic grids. At this point, it is the low-resolution data that is being used in the simulation. The hotspot appears to be substantially attenuated

when using the distance-to-shore parameter probably due to particles crossing the boundary in a less focussed point over a stretch of the coastline. The reason for the hotspot is out of the scope of the analysis but it is nevertheless surprising given the distance from the release points.

- As expected, the distance-to-shore scenario does correctly show a hotspot (>2500 particles) around the Llobregat River release point given the geometry and hydrodynamic conditions. This does not seem to register when using the hydrodynamic grids as boundary detection showing that the complex geometry of that area that is adjacent to the Barcelona port is better resolved when using the distance-to-shore parameter, thus having a visible effect on where particles become beached.
- Likewise, there are several other hotspots within the high-resolution areas and outside of these which are picked up by the distance-to-shore beaching scenario and not by the hydrodynamic grid boundary detection scenario.
- From the two density maps, it would appear that when the distance-to-shore parameter is used the resulting densities along the shoreline align more closely to what would be expected from the area at such a scale, including an important hotspot at the main release point, which would justify recommending using the distance-to-shore parameter along with high-resolution hydrodynamic data at localised scales.
- Beaching density maps were not included because numerically it was seen that the majority of beaching occurred around the Llobregat river mouth for both nested grids and the CMEMS only grid, and this could obscure any beaching patterns diagrammatically speaking. The beaching patterns of deposition along the coastline based on the distance were deemed more useful.

It is of course entirely up to the authors how they wish to compare these beaching scenarios, but I do not see how the manuscript, in its current form, can make a justified recommendation about which scenario is 'best'.

- Given the above example, the reduction in beaching artefacts, the independence from the resolution of the hydrodynamic grids across the domain where varying resolutions are used, the low risk of missing coastal processes where the low-resolution grid applies that does not resolve these processes anyway, we believe that the recommendation for the use of a distance-to-shore approach to beaching for small scale studies is justified.
- In the revised manuscript in the discussion, we replaced lines 551-559 with:
  A distinct approach to particle beaching was provided in scenario 2 which introduced a deterministic beaching model that relied on a physical shoreline and pre-calculated distance data of nodes to the shoreline in a grid(s) used in a fieldset. This way, the distance between particles and the shoreline could be calculated during the simulation to determine when and where they cross the land-water boundary as illustrated in the beaching sensitivity analysis Fig.Xc and Fig.Xd. Furthermore, the distance-to-shore beaching module is independent of the hydrodynamic data resolution. This parameterisation effectively tells a particle the minimum distance to the predetermined shoreline in real-time, and if the distance to the shore is <0 the particle becomes beached. This method can offer a more precise detection of particle beaching along the shoreline at smaller coastal scales. Using the distance to the shoreline and a physical boundary detection ensures consistency throughout the study domain, even when nesting grids of varying resolutions. Additionally, the interpolation and grid nesting capabilities of Parcels allowed distance calculations to not

be limited by a decrease in spatial resolution throughout the domain. Shoreline detection based on a physical boundary does have some limitations, such as relying on the availability of high-resolution spatial data and requiring preprocessing steps. While obtaining detailed coastline data may not be feasible for larger-scale studies, the advantages it offers for smaller-scale studies are substantial, particularly when assessing the impact of beaching at a localised level.

- In the conclusion we state:
  The LOCATE model effectively integrated high-resolution hydrodynamic data around areas of high interest and used high-resolution shoreline data, providing greater confidence and precision in the detection of the land-water boundary and particle accumulation zones, which becomes more salient the smaller the scale of the study.

The other clarification I wanted to make was on the diffusive parameterisation. I did not intend to question the use of kh $=10$ m$^2$s$^{-1}$ for the IBI-CMEMS grid. My question is why the same value of Kh was used for the finer (coastal and harbour) grids, which should have a much lower value of Kh, or none at all?

- We understand that the dispersion process depends on the size of the mesh and resolution used, and in theory should be variable depending on the resolution of the data used and also on the velocities therein. In practice, however, we do not have experimental or empirical data to know for sure what value of Kh to use at these scales. We therefore thought it best to be prudent to this respect and use the same Kh value throughout the study domain for consistency while recognising that this is an area of future research. In November 2023 we finished a drifter campaign where we released a series of drifters in pairs and groups from selected areas close to the Barcelona coastline (such as Bogatell beach or the Llobregat river mouth) where we will try to validate the parameters of dispersion in this area and with the scales used in this study. Nevertheless, the validation of the drifter data for this work showed good skill score values, given that these trajectories were within where the high-resolution data applied and that a Kh value of 10 m$^2$s$^{-1}$ was used.

---

## Referee Report (RR1)

**Comments on revised manuscript**

I thank the authors for their comprehensive responses to my comments on their manuscript. The revised manuscript has addressed most of my concerns, and the purpose of the manuscript is now clear (and convincing). I continue to disagree with the interpretation of the perceived benefits of scenario 2 over scenario 1, but consider the results to be robust and would therefore recommend this manuscript for publication, pending minor revisions. In particular, I ask the authors to acknowledge the potential for scenario 2 to introduce artefacts due to the inconsistency between the hydrodynamic coastline, and the coastline used in the beaching parameterisation.

**Major comments**

In their response, the authors state that the main purpose of the beaching parameterisation is "to predict where beaching occurs along the coastline with greater certainty and precision". I agree. This being the case, the authors make the following arguments in support of their favourable interpretation of scenario 2:

1. The hydrodynamic grids do not resolve key geometric features of the coastline such as piers and groynes, which may significantly affect beaching patterns. A beaching parameterisation that accounts for these unresolved features using high-resolution coastline data therefore improves the accuracy of beaching distribution predictions, particularly for small (beach-scale) studies.
2. Using the hydrodynamic grid to infer beaching generates artefacts, such as at domain boundaries and grid cell corners.

**Point 1 – unresolved features**

On pages 20-21 of their response, the authors demonstrate how scenario 1 is unable to resolve beaching on many of the fine-scale features of the Barcelona city area, such as groynes and piers. In contrast, the pattern of beaching in scenario 2 (pages 22-23) resolves all these features.

Superficially, the figures on pages 22-23 look like a significant improvement. The problem is that figures 22-23 are based on a coastline and hydrodynamics that are physically inconsistent, and I therefore do not think it is obvious that this improvement is 'real'.

From the figure below (page 21), we can clearly see that these piers are not resolved by the hydrodynamic model. The hydrodynamic model therefore (obviously) assumes these piers do not exist, and simulates currents that can straight through them. If the hydrodynamic model resolved these piers, this would not be the case – currents would be deflected around the piers. Of course, beaching could still occur on these piers if the currents were deflected (e.g. by Stokes drift, wind, getting stuck in rocks, etc) but these are different from the process driving beaching in scenario 2 simulations, which is artificial convergence due to mismatch between the coastlines assumed by the hydrodynamics, and by the beaching parameterisation. Scenario 2 would predict a trajectory like the red arrow in the figure below. A particle moves parallel to the model coast, attempts to pass through a pier (because the hydrodynamic model isn't aware of it), and then 'beaches' as it reaches the pier. Instead, it is entirely possible that the 'true' current is deflected around the pier, and that the particle doesn't beach. Without resolving the pier in the hydrodynamic model and/or having data for validation, we do not know which one is true.

[Figure]

Fundamentally, we have no useful data beyond the resolution of the hydrodynamic grid. Figures 22-23 look nice and neat, but they're not based on physics. Attempting to resolve sub-grid scale features might make the results look more realistic, but that doesn't mean they necessarily are more realistic.

The authors state that "having a model that predicts that debris becomes beached several kilometres inland renders it useless in knowing which beaches in or around the city would be more affected by debris". This is correct, but imposing a high-resolution coastline isn't necessarily going to change this, since we are not resolving the relevant physics. This is why many studies pass particle distributions through a Gaussian filter with a standard deviation several times greater than the model resolution, in recognition of the fact that a model cannot meaningfully resolve structures at or below grid resolution (e.g. Mitarai et al., 2009).

Maybe scenario 2 does improve the accuracy of predictions – perhaps the inconsistency between the hydrodynamic and parameterised coastline isn't a big problem. However, unless the authors have evidence supporting this (e.g. in the form of fine-scale debris observations), we have no way of knowing this.

**Point 2 – artefacts**

This second point is interesting, and the particle density plots on page 26 of the authors' response is particularly useful – I would recommend adding this to the supplementary materials. Although I agree that scenario 2 appears to reduce some artefacts (particularly the cell-corner and domain boundary 'hotspots'), for the same reasons as explained in point 1, how do the authors know that it hasn't introduced new artefacts? As an example, from this figure, we can see that the model coastline is inland of the true coastline, and based on the particle distribution, we can guess that there is a westward coastal current. Ignoring Stokes drift, the flow should be broadly parallel to the coast (arrow below). And indeed, in scenario 1, we therefore see very low particle density along the coast here.

[Figure]

However, with scenario 2, particles will beach as they intercept the 'coastline' as they travel westwards. This convergence with the coastline is not real. If the hydrodynamic model was aware of the real coastline, the currents would be different. Indeed, the accumulation hotspot indicated by the authors in the figure below is entirely consistent with this hypothesis of artificial convergence with the 'true' coastline. The higher beaching amount under scenario 2 (Table 4) is also consistent with this, due to this artificial convergence (and therefore beaching). Without in-situ data for comparison, there is no way of knowing whether the prediction of accumulation around 1.5E, 41.17N is an improvement over the Scenario 1, or entirely artificial.

[Figure]

*Suggestions*

In summary, I think the comparison between scenarios 1 and 2 is interesting and publishable. My concern is that the manuscript argues that scenario 2 is an *improvement*, and I cannot see any evidence supporting this claim, beyond the elimination of certain artefacts. I would recommend that the authors remove claims that scenario 2 improves the 'precision' (and certainly 'accuracy') of predictions or, at the very least, acknowledge that the scenario 2 parameterisation is inconsistent with the hydrodynamic coastline, and therefore runs the risk of introducing new artefacts (a detailed investigation of which is beyond the scope of the present study). The paragraph starting on line 510 and, in particular, the sentence starting on line 612, may be an appropriate place to mention this inconsistency, and that the fine-scale flow around groynes, piers etc. will also not be resolved.

Lines of concern that I would strongly recommend removing or revising are as follows:

- L17: "…represent deposition patterns with greater precision of particle beaching locations using high-resolution shoreline data"

- L600-605: "…but to provide a more reasonable prediction of where beaching can occur with greater certainty and precision, especially at coastal scales. In practical terms, employing a distance-to-shore parameterisation and high-resolution hydrodynamic data could be more effective at identifying which beaches around the Barcelona metropolitan area could be more impacted by a discharge event after heavy rainfall, where small-scale structures were resolved as seen in Fig.8h. Other scenarios do not resolve structures at small scales, making the quantification of beaching at specific locations more difficult."
- Line 638: "…and particle accumulation zones"

**Minor comments**

- I am still not sure what the point of equations 3 is, given that this study assumes $K$ is a constant. I would recommend just giving equation 4, whilst acknowledging that this is a simplification.
- From the authors' response, it does not appear that an understanding of the temporal variability in debris input is relevant to the interpretation of results in the manuscript. I would suggest moving Figure 4 to the supplementary materials for brevity.
- Concerning Figure 8, the authors wrote in their response that they would use a divergent colourmap. However, the figure still uses a sequential colourmap (e.g. see the cmocean or cmasher packages).
- Concerning Figure 9, the authors state in their response: "and while there may be particles showing blue dots on the left side plots, the concentrations may be low and not be enough to show on the heatmaps". However, the colour bar for the concentration maps start at 0, indicating that all cells with at least one particle ($> 0$ km$^{-2}$) should be coloured.
- I would recommend adding a line at x = 1 to the rightmost panel of Figure 10 (since this is equivalent to both grids being the same. Alternatively, consider changing the x axis to a logarithmic scale.

**Technical comments**

- Line 408: "Additionally, small-scale structures, such as piers and groynes do not seem to be considered" – I would suggest changing 'do not seem to be' to 'are not'.

**References**

Mitarai, S., Siegel, D. A., Watson, J. R., Dong, C., & McWilliams, J. C. (2009). Quantifying connectivity in the coastal ocean with application to the Southern California Bight. *Journal of Geophysical Research: Oceans*, *114*(10), 1–21. https://doi.org/10.1029/2008JC005166

---

## Author Response (AR2)

**Author's reply**

Firstly, we wish to thank Referee 1 for the extensive explanations of their arguments, which are very valid and convincing, it is much appreciated. While the authors feel that using the distance grid in the beaching parameterisation for scenario 2 provides clearer results, the claims of increased precision and improvements have been removed, because as Referee 1 points out, more data would be required to objectively substantiate the claims, which is beyond the scope of this article. Also as Referee 1 mentioned, the inconsistencies between the hydrodynamic and the real coastlines are now highlighted in the manuscript. We do state, however, that in the absence of such high-resolution hydrodynamic data that can resolve individual piers, groynes and other small-scale structures the distance-to-shore parameterisation can provide a suitable compromise when looking at beaching patterns and comparing to using hydrodynamic data alone where such structures are not visible, as well as providing uniformity of the coastline when using varying resolutions with nested grids. We also acknowledge and mention the possibility that using the distance-to-shore parameterisation can introduce artificial convergence artefacts and include a figure in the Appendix as suggested by Referee 1 to this effect. We believe we have now addressed and accommodated all of the concerns that Referee 1 has highlighted and wish to thank them for their thoroughness and detailed approach, and for providing such well-rounded arguments that have enriched this manuscript.

Replies to the individual comments are in blue. The concerns in the referee's major comments are worked into the replies below.

**Suggestions**

In summary, I think the comparison between scenarios 1 and 2 is interesting and publishable. My concern is that the manuscript argues that scenario 2 is an *improvement*, and I cannot see any evidence supporting this claim, beyond the elimination of certain artefacts. I would recommend that the authors remove claims that scenario 2 improves the 'precision' (and certainly 'accuracy') of predictions or, at the very least, acknowledge that the scenario 2 parameterisation is inconsistent with the hydrodynamic coastline, and therefore runs the risk of introducing new artefacts (a detailed investigation of which is beyond the scope of the present study). The paragraph starting on line 510 and, in particular, the sentence starting on line 612, may be an appropriate place to mention this inconsistency, and that the fine-scale flow around groynes, piers etc. will also not be resolved.

From the concerns and recommendations, the changes proposed by Referee 1 are included in the discussion and conclusion. The paragraph starting on line 510 deals with the resolutions in the hydrodynamic data, and a more suitable place for the inclusion of these points is further on, in the last two paragraphs of the discussion.

Lines of concern that I would strongly recommend removing or revising are as follows:

- L17: "…represent deposition patterns with greater precision of particle beaching locations using high-resolution shoreline data"

Lines 15 to 18 in the abstract are changed to:

LOCATE can effectively integrate high-resolution hydrodynamic data within nested grids to model the dispersion and deposition patterns of particles at coastal scales using high-resolution shoreline data for shoreline detection uniformity.

- L600-605: "…but to provide a more reasonable prediction of where beaching can occur with greater certainty and precision, especially at coastal scales. In practical terms, employing a distance-to-shore parameterisation and high-resolution hydrodynamic data could be more effective at identifying which beaches around the Barcelona metropolitan area could be more impacted by a discharge event after heavy rainfall, where small-scale structures were resolved as seen in Fig.8h. Other scenarios do not resolve structures at small scales, making the quantification of beaching at specific locations more difficult."

The last two paragraphs of the discussion with lines 598 to 614 in the previous manuscript have been changed to:

The main purpose of this beaching parameterisation is not to predict that beaching occurs on the real shoreline, but to provide a consistent coastline independently of the hydrodynamic resolution used when nesting grids. Additionally, the interpolation and grid nesting capabilities of Parcels allowed distance calculations not to be limited by a decrease in spatial resolution throughout the domain. Although small-scale structures are seemingly resolved using this parameterisation allowing for quantification of beaching at specific locations with much less difficulty than other scenarios, it is not consistent with the hydrodynamic coastline. Therefore, the flow around the sub-grid scale features resolved may not be based on physical processes and the localised effects these structures could have on the hydrodynamic data are not considered. Additionally, the potential for the introduction of artefacts from artificial convergence cannot be ruled out in areas where the hydrodynamic coastline and the real coastline based on high-resolution shoreline data converge. Whether these inconsistencies have material effects on the prediction of beaching patterns remains an area for future work. Other limitations of this scenario include the dependency on the availability of high-resolution spatial data and the requirement of preprocessing steps. In the absence of hydrodynamic data of such a fine resolution that may counter these shortfalls, this beaching parameterisation can provide a suitable compromise for small-scale studies and could lead to the development of further parameterisations at beach level in future research. It is crucial to underscore that the considerations for using a distance-to-shore beaching parameterisation are especially relevant for small-scale or localised studies where stakeholders may prioritise identifying specific at-risk areas. In contrast, concerns at a larger scale may differ significantly and the parameterisations used in scenario 2 may not be as useful or meaningful then.

- Line 638: "…and particle accumulation zones"
Lines 636-639 in the previous manuscript have been changed from:
Despite these constraints, the LOCATE model effectively integrated high-resolution hydrodynamic data using nested grids around areas of high interest and used high-resolution shoreline data, providing greater confidence and precision in the detection of the land-water boundary and particle accumulation zones, which becomes more salient the smaller the scale of the study

Changed to:

Despite these constraints, the LOCATE model effectively integrated high-resolution hydrodynamic data using nested grids around areas of high interest and used high-resolution shoreline data to provide land-water boundary detection uniformity throughout the domain when using varying hydrodynamic resolutions.

- I am still not sure what the point of equations 3 is, given that this study assumes K is a constant. I would recommend just giving equation 4, whilst acknowledging that this is a simplification.

The authors insist on keeping the general equation 3 for context as is. Although the K value is constant in the study due to no other data being available as discussed in the previous round of comments, should this change then it would be relevant to see how equation 3 is simplified to equation 4. Given that so many other points suggested have been included, the authors wish to keep this one as is.

- From the authors' response, it does not appear that an understanding of the temporal variability in debris input is relevant to the interpretation of results in the manuscript. I would suggest moving Figure 4 to the supplementary materials for brevity.
As per this suggestion, the figure has been moved to the Appendix

- Concerning Figure 8, the authors wrote in their response that they would use a divergent colourmap. However, the figure still uses a sequential colourmap (e.g. see the cmocean or cmasher packages).
Apologies for this oversight in stating that, this was tried but did not work well, and I forgot to change the comment. For a divergent map, the lightest shade should be at 0, provided the scale would diverge similarly from that point. However, the scale goes from -10 to 2, with the midpoint at around -4 which doesn't make much sense, and having a light shade at 0 offers very little contrast with the background. The authors feel the figure is fine as is using the viridis colourscale which shows the distance range appropriately.

- Concerning Figure 9, the authors state in their response: "and while there may be particles showing blue dots on the left side plots, the concentrations may be low and not be enough to show on the heatmaps". However, the colour bar for the concentration maps start at 0, indicating that all cells with at least one particle ($> 0$ km$^{-2}$) should be coloured.
This figure now shows the concentrations starting at 1 per km$^2$. Also, in the previous round of revision, Referee 1 commented on the simulation snapshot for 31 May 2017. Upon double checking (again), the referee was correct in that the maps did not align, and has now been replaced with the correct day that now corresponds to the concentration heatmap. This was due to an oversight in zero indexing of days from the beginning of the month when producing the snapshots.

- I would recommend adding a line at x = 1 to the rightmost panel of Figure 10 (since this is equivalent to both grids being the same. Alternatively, consider changing the x axis to a logarithmic scale.
The x-axis of Fig 9c (formerly Fig 10) is now on a logarithmic scale and is easier to interpret. With thanks to Referee 1 for the suggestion.

- Line 408: "Additionally, small-scale structures, such as piers and groynes do not seem to be

considered" – I would suggest changing 'do not seem to be' to 'are not'.

This has been changed.